# Plausible pathway for a host-parasite molecular replication network to increase its complexity through Darwinian evolution

**Rikuto Kamiura**[1], **Ryo Mizuuchi**[2,3], **Norikazu Ichihashi**[1,3,4]*

**1** Department of Life Science, Graduate School of Arts and Science, The University of Tokyo, Tokyo, Japan,
**2** JST, PRESTO, Kawaguchi, Saitama, Japan, **3** Komaba Institute for Science, The University of Tokyo,
Tokyo, Japan, **4** Research Center for Complex Systems Biology, Universal Biology Institute, The University
of Tokyo, Tokyo, Japan

* ichihashi@bio.c.tokyo-u.ac.jp

Center for Theoretical Physics - South American
Institute for Fundamental Research, BRAZIL

**Data Availability Statement:** All relevant data are
within the manuscript and its Supporting
Information files.

## Abstract

How the complexity of primitive self-replication molecules develops through Darwinian evolution remains a mystery with regards to the origin of life. Theoretical studies have proposed that coevolution with parasitic replicators increases network complexity by inducing interdependent replication. Particularly, Takeuchi and Hogeweg proposed a complexification process of replicator networks by successive appearance of a parasitic replicator followed by the addition of a new host replicator that is resistant to the parasitic replicator. However, the feasibility of such complexification with biologically relevant molecules is still unknown owing to the lack of an experimental model. Here, we investigated the plausible complexification pathway of host-parasite replicators using both an experimental host-parasite RNA replication system and a theoretical model based on the experimental system. We first analyzed the parameter space that allows for sustainable replication in various replication networks ranging from a single molecule to three-member networks using computer simulation. The analysis shows that the most plausible complexification pathway from a single host replicator is the addition of a parasitic replicator, followed by the addition of a new host replicator that is resistant to the parasite, consistent with the previous study by Takeuchi and Hogeweg. We also provide evidence that the pathway actually occurred in our previous evolutionary experiment. These results provide experimental evidence that a population of a single replicator spontaneously evolves into multi-replicator networks through coevolution with parasitic replicators.

## Author summary

How primitive simple self-replication molecules develop their complexity through evolution is one of the largest mysteries in the origin of life. The largest obstacle in the development of complexity is parasitic replicators, which spontaneously appear and destroy intermolecular cooperative networks, such as hypercycles, and simplify the replication system. However, Takeuchi and Hogeweg found that parasitic replicators could increase the

**Funding:** This work was supported by Japan Science and Technology Agency, CREST grant number JPMJCR20S1 (N.I.) (https://www.jst.go.jp/kisoken/crest/) and Japan Society of Promotion of Science, KAKENHI grant number 20H04859 (N.I.) (https://kaken.nii.ac.jp/). The funders had no role in study design, data collection and analysis, decision to publish, or preparation of the manuscript.

**Competing interests:** The authors have declared that no competing interests exist.

complexity of replication network by working as a "niche" for multiple host replicators. This idea provides an attractive answer to the long-standing mystery, that is, how complexity of a molecular replication system develops, although experimental evidence is lacking. In the present study, we performed a theoretical analysis of an RNA replication system using computer simulation, together with experimental verification, to understand the reason for sustainable co-replication of multiple replicators. We also found that the most plausible route for complexity in the host–parasite replication network is the addition of the parasite first, followed by a new host that is resistant to the parasite. These results provide both theoretical and experimental evidence that parasitic replicators mediate the development of complexity in replication networks through Darwinian evolution.

## Introduction

Most of the origin of life scenarios hypothesize that a simple self-replicating molecule or a set of replicating molecules appeared and underwent Darwinian evolution to gradually become more complex toward the extant life [1–5]. To examine the plausibility of this scenario, researchers have synthesized self-replication molecules such as simple RNA or peptides, which might have been available on the early Earth (reviewed in [4–7]), although Darwinian evolution of these simple molecules remains a challenge. RNA or DNA replication systems capable of Darwinian evolution have been constructed using proteins of the existing organisms [8–11]. Although the proteins used in these systems did not exist on the early Earth, they could be utilized as experimental models that might mimic some aspects of primitive replicators consisting of biologically relevant molecules such as RNA and peptides. Even for replication systems consisting of modern proteins, however, the development of complexity through Darwinian evolution, a prerequisite for the emergence of life, remains a significant challenge.

Complexity is an ambiguous concept, and there are several measures for determining the complexity of a replication system, such as the amount of information encoded in a replicator [12], the number of traits of a replicator [13], the difficulty in achieving traits [14], the potential to cope with environmental uncertainty [15] and the number of replicators organized as a replication network [16, 17]. Here, we focus on one of the measures, the number of replicators in a replication network (i.e., network complexity). One of the possible pathways for a replicator to develop this complexity is the diversification of replicators and formation of inter-dependent, cooperative replication networks among them, such as a hypercycle [2, 18–21]. A major hurdle for inter-dependent network formation is parasitic replicators, which destroy the cooperative replication network [18, 22]. To date, theoretical [23–27] and experimental [28, 29] studies have revealed that spatially restricted systems, such as compartmentalization, repress parasitic replicators. Furthermore, Takeuchi and Hogeweg theoretically demonstrated that parasitic replicators induced diversification of RNA-like replicators through an evolutionary arms race in two-dimensional square grid and allowed the formation of more complex inter-dependent molecular networks [16]. In their simulation, an RNA population evolved ecological complexity by successive appearance of a parasitic RNA, followed by the addition of an RNA that is resistant to the parasitic RNA into the replicator network. The study suggests that parasitic replicators, which have been considered as an obstacle to complexification, may play an important role in complexification in compartmentalized structures. One of the remaining challenges is the plausibility of such complexification within a realistic parameter space that is achievable with biologically relevant molecules such as RNA and proteins.

Recently, we constructed an in vitro translation-coupled RNA replication system and demonstrated the coevolution of host and parasitic RNAs [29, 30]. In this system, a host RNA replicates through the translation of the self-encoded replication enzyme, whereas parasitic RNAs, which spontaneously appear, replicate by relying on the replication enzyme translated from the host RNAs. When the replication was repeated through serial replication processes in water-in-oil compartments, the RNAs were mutated by replication errors and underwent Darwinian evolution. In a previous study, we showed that host and parasitic RNAs diversified into multiple lineages through Darwinian evolution [30]. In a recent study, we further repeated the serial replication processes and found that the diversified RNA species start to co-replicate by forming an inter-dependent network, which finally consists of three hosts and two parasites [31]. These experimental results support the idea that coevolution between host and parasitic replicators can drive diversification and complexification. However, it is still unknown how such complexification is possible and competitive exclusion among RNA species is circumvented.

According to the previous theoretical study by Takeuchi and Hogeweg [16], such complexification is explained by the successive evolution of a parasitic species, which function as a niche for a new host species. This study examined whether our experiment can support the complexification process. To this end, we first constructed a theoretical model of a compartmentalized host-parasite replication system, which is conceptually the same as that developed by Takeuchi and Hogeweg but formatted to be similar to our experimental system. We then investigated the parameter space that allows for the sustainable replication of all replicators in each host-parasite replication network in up to three-member networks using computer simulation. We also conducted an evolutionary simulation by introducing new replicators with different parameters. These simulations were consistent with those of the previous study showing that the most plausible complexification pathway of a single host involves the successive addition of a parasite first and then a new host that is resistant to the parasite. Furthermore, we proposed that the most plausible pathway (the addition of a parasite, followed by the addition of a parasite-resistant host) occurred in our previous evolutionary experiment.

## Results

### Strategy of theoretical model and analysis

To investigate the parameter space that allows the sustainable replication of all members in a replication network, we constructed a theoretical model of compartmentalized host–parasite replication. In this model, the compartmentalized replication cycle consists of repeating three steps: replication, culling, and fusion-division (Fig 1). The detailed procedures are described in the Methods section.

The model constructed in this study, which mimics our previous evolutionary experiments [29], differs from those of previous studies in the literature at some points. For example, some replicator models assume compartments aligned in 2D space, where each compartment interacts with only adjacent compartments [16, 32], but our compartments are well-mixed and can fuse with any compartment. In other studies, the cellular compartments are assumed to grow and divide depending on the internal reaction [23, 24, 33], whereas in our model, the volumes of the compartments and the fusion-division steps are independent of the internal reaction.

Using this model, we investigated a possible complexification pathway for host-parasite replication network. Fig 2A shows a possible complexification pathway with up to three members. A single host replicator, termed "H," possibly forms two-member replication networks by the addition of a new host or parasite, termed HH and HP networks, respectively. The next step is to form the three-member networks, named HHH, HHP, and HPP. In this study, we

## A

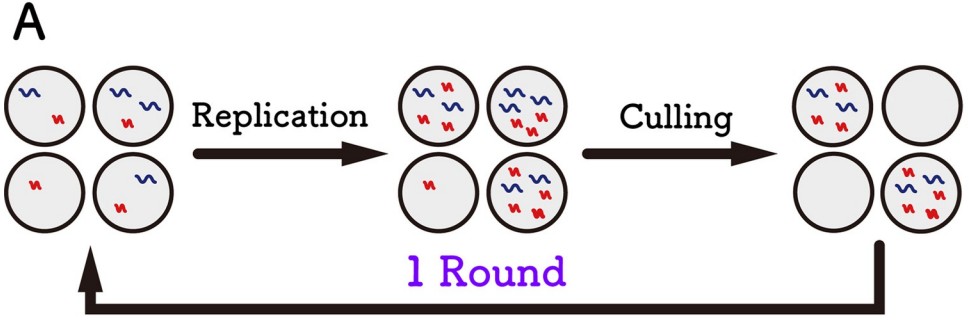

## B (Replication)

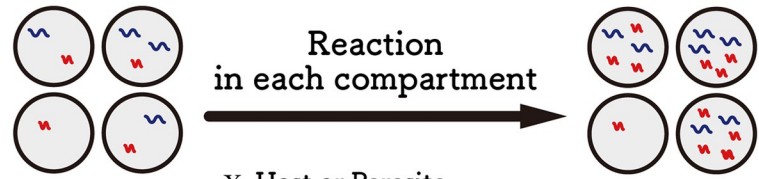

$X$: Host or Parasite

$$\frac{\mathrm{d}X_i}{\mathrm{d}t} = X_i \left( \sum_j k_{ji}^{\mathrm{X}} X_j \right) \left( 1 - \frac{\sum_j X_j}{N} \right)$$

$X_i$ : number of replicator i
$H_i$ : number of host i
$k_{ji}^{\mathrm{X}}$ : replication coefficient
 of host j to Xi
$N$ : carrying capacity

## C (Culling)

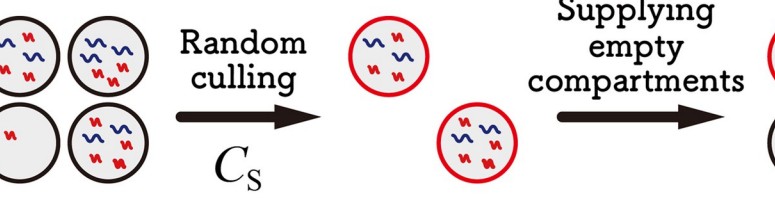

$C$ : total number of compartments
$C_{\mathrm{s}}$ : number of selected compartments
$S$ : culling rate

$$C_{\mathrm{S}} = \lfloor C \cdot S \rfloor$$

## D (Fusion & Division)

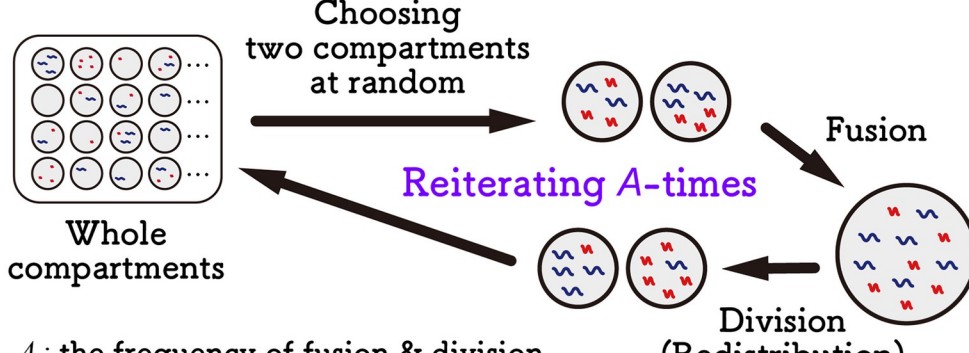

$A$ : the frequency of fusion & division

**Fig 1. Theoretical model of compartmentalized replication through serial replication cycles.** (A) Overview of the serial replication cycle of compartmentalized replication, which consists of replication, culling, and fusion-division steps. (B) In the replication step, hosts and parasites in each compartment replicate according to differential equations. (C) In the culling step, a certain number (Cs) of compartments are randomly selected, and the other compartments are replaced with empty compartments. (D) In the fusion-division step, two compartments are randomly chosen, and the internal host and parasites are mixed, followed by random redistribution into two compartments. These processes were repeated $A$ times. The number of compartments is 3,000, and the frequency of fusion-division is 5,000 unless indicated otherwise.

evaluated the stability of each network by running a simulation with various parameters for certain time steps (rounds) and measuring how many times all members in the network existed in the last time step. The parameters used here are replication coefficients, with which each host replicates itself or other replicators. For example, the HP network can be characterized by two replication coefficients, that for self-replication ($k_{11}^{H}$) and that for the replication of the parasite ($k_{11}^{P}$) (Fig 2B).

## HH networks

First, we investigated the HH network, in which two host species (Host 1 and Host 2) self-replicate with coefficients $k_{11}^{H}$ and $k_{22}^{H}$ and cross-replicate with coefficients $k_{12}^{H}$ and $k_{21}^{H}$, respectively (Fig 3A). We performed computer simulations of the compartmentalized replication shown in Fig 1 with all combinations of four values (1.7, 2.0, 2.3, and 2.6) for each coefficient. The four values are based on experimental data that were obtained in the later experiment conducted in this study (Table 1, estimated in the "Parameter estimation of the representative RNAs" section below), in which the maximum and minimum coefficients were approximately 2.3 and 2.0, respectively, and additionally we chose a larger value (2.6) and a smaller value (1.7). We also tested more extreme values (0.2 and 4.1) in S1 Fig.

Using the combinations of these coefficients, we performed 100 independent simulations and counted the number of "sustained" runs in which all replicators were sustained for 100 rounds of the serial replication cycle (Fig 3B). With most of the parameter sets, the HH replication network was not sustained even once, suggesting that the co-replication of two types of host species is unlikely. However, there are some parameter sets that allow sustained replications for all 100 runs, which can be categorized into two conditions. Condition I is a "low self- and high cross-replications" condition (i.e., $k_{12}^{H}, k_{21}^{H} > k_{11}^{H}, k_{22}^{H}$, Fig 3C), where the two hosts replicate cooperatively, marked with red squares in Fig 3B. Condition II is a "balanced

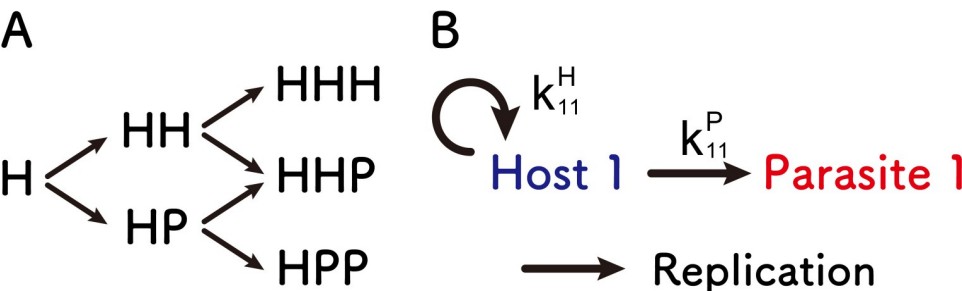

**Fig 2. Possible complexification pathways and an example of replication parameters.** (A) Possible complexification pathways in up to three-member replication networks. Starting from a single host self-replicator (H), next possible steps are the addition of another host or parasite to form two-member replication networks, namely, HH and HP. In the next step, another host or parasite could join to form three-member replication networks, namely, HHH, HHP, and HPP. (B) Parameters that characterize HP network as an example. A host replicates itself (i.e., self-replicates) with the coefficient $k_{11}^{H}$ and a parasite with the coefficient $k_{11}^{P}$. Similar coefficients are used for the other networks.

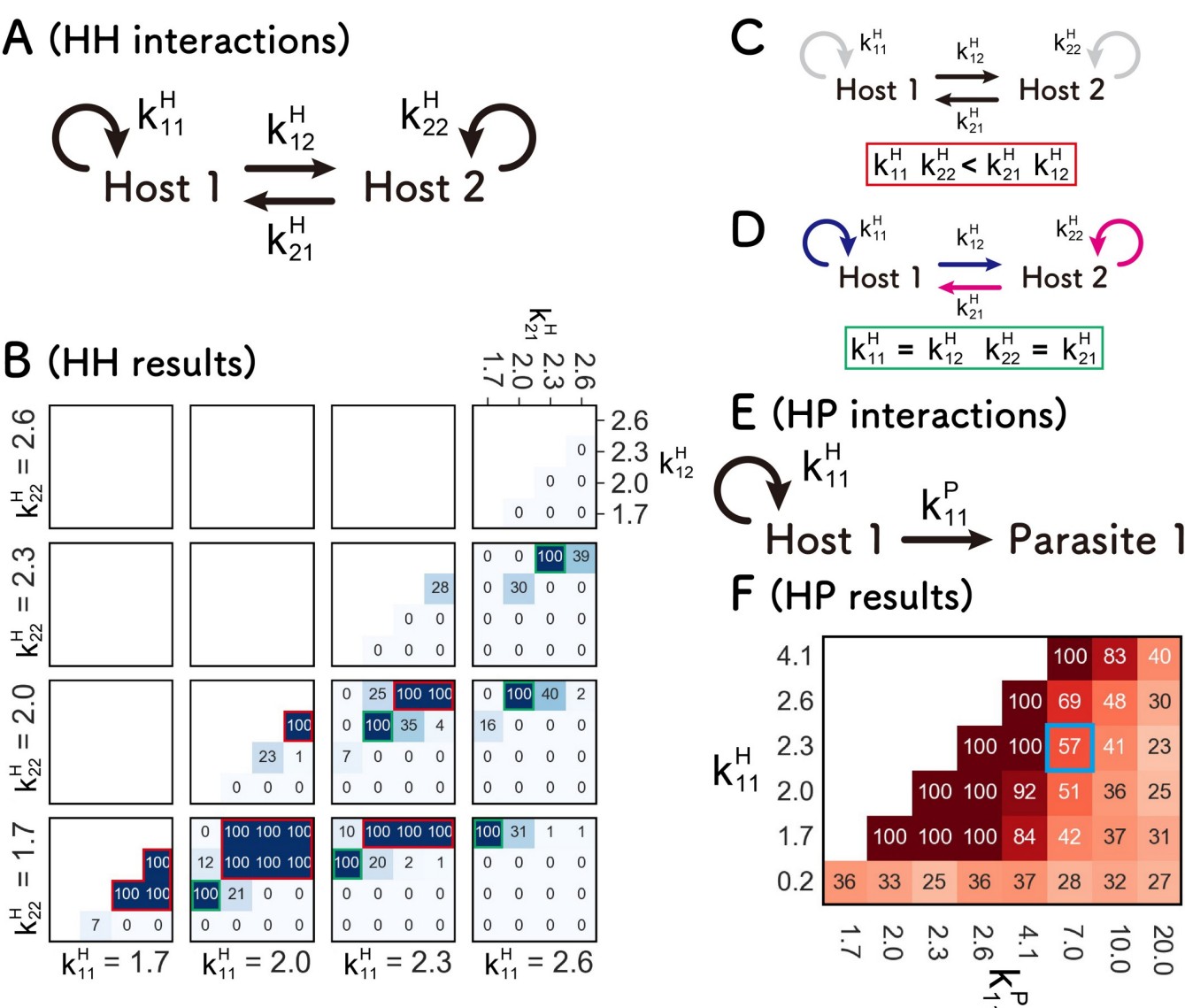

**Fig 3. Search for the parameters that allow sustainable HH and HP networks.** (A) Scheme of the HH network. Each host self-replicates with coefficient $k_{11}^H$ or $k_{22}^H$, and replicates the other host with coefficients $k_{12}^H$ or $k_{21}^H$. (B) Numbers of the runs in which both Hosts 1 and 2 are sustained for 100 rounds out of 100 independent simulations. The regions enclosed with red and green squares are the two different sustainable conditions each depicted in (C) and (D), respectively. The results on the diagonal line were omitted because Hosts 1 and 2 are identical there. (E) Scheme of the HP network. The host self-replicates with coefficient, $k_{11}^H$, and replicate the parasite with coefficient, $k_{11}^P$. (F) Numbers of the runs in which both the host and parasite are sustained for 100 rounds out of 100 independent simulations. The blue square indicates parameters of the dominant RNAs (Host$^1_{exp}$ and Parasite$^1_{exp}$) obtained in the evolutionary experiment.

**Table 1. Estimated replication coefficients.**

| | | Replicating RNA | | |
| --- | --- | --- | --- | --- |
| | | Host$^1_{exp}$ | Host$^2_{exp}$ | Parasite$^1_{exp}$ |
| Replicated RNA | Host$^1_{exp}$ | 2.3 ($k_{11}^H$) | 2.3 ($k_{21}^H$) | 0.0 |
| | Host$^2_{exp}$ | 2.3 ($k_{12}^H$) | 2.0 ($k_{22}^H$) | 0.0 |
| | Parasite$^1_{exp}$ | 6.7 ($k_{11}^P$) | 0.0 ($k_{21}^P$) | 0.0 |

*Corresponding parameter names in the theoretical model are shown in parentheses.

replication" condition ($k_{11}^H = k_{12}^H, k_{21}^H = k_{22}^H$), Fig 3D, where one host species self-replicates as much as cross-replicates with the other host species, marked with green squares. These results indicate that sustainable replication in the HH network occurs only in limited cases. The results are similar for the extreme parameters (S1 Fig). Notably, the "balanced replication" condition, which requires the identical parameters, is unlikely with realistic molecules. It should be noted that the sustainability evaluated here was limited to up to 100 rounds and longer sustainability was not guaranteed. This is also the case for the following simulations.

## HP networks

Next, we investigated the HP network, in which a host self-replicates with coefficients $k_{11}^H$ and replicates a parasite with coefficients $k_{11}^P$ (Fig 3E). For these replication coefficients, we used the same values as those used in the HH network (Table 1, estimated in the "Parameter estimation of the representative RNAs" section below), including the two extreme values (0.2, 1,7, 2.0, 2.3, 2.6, and 4.1). Additionally, we adopted the experimental value (7.0) and larger values (10.0 and 20.0) for the coefficients of the parasite ($k^P$). Using the combinations of these coefficients, we performed a series of computer simulations and counted the number of sustained runs out of 100 independent runs (Fig 3F). The number of sustained runs gradually changed within the parameter space. This result exhibits a clear contrast to the HH network in Fig 3B, in which the sustained parameter was relatively rare, and the number of sustained runs changed sharply with a small parameter change. These results indicate that the HP network is more sustainable in a broader parameter space than the HH network.

The number of sustained runs may depend on the number of compartments. To test this hypothesis, we conducted simulations with a varied number of compartments, *C*. The frequency of fusion-division, *A*, was also changed proportionally to *C* to keep the average fusion-division number per compartment constant. In the HP network (S2A Fig), the number of sustainable replications increased when the number of compartments was increased from 3,000 to 10,000 (the original number was 3,000). This is probably because a large number of compartments provide hosts with a greater chance of escaping from parasites. Similarly, in the HH network (S2B Fig), the parameter sets that allow sustainable replication slightly increased when the number of compartments was increased from 3000 to 10,000 (compare Fig 3B with S2B Fig, left and S1 Fig with S2B Fig, right), while the two hosts were still unsustainable, with most of the parameter sets. These results confirm that the HP network is sustainable in a broader parameter space than the HH network, even with a larger number of compartments.

## HPP network

Next, we investigated the HPP network, in which another parasite was added to the HP network, with the same simulation method and parameters as for the HP network (Fig 4A). The number of sustained runs, in which all host and parasites are sustainably replicated for 100 rounds, in 100 independent simulations, are shown in Fig 4B. The host and two parasites are never sustainably replicated together with any combinations of parameters used here, mainly due to the competition between the two parasites, which immediately exterminates one with a smaller coefficient. Even after increasing the number of compartments to 10,000, no sustainable replication was observed (S2C Fig). These results suggest that even if an HPP network is formed during evolution, it will soon return to the HP network.

## HHP network

Next, we investigated the HHP network, in which another host was added to the HP network or another parasite was added to the HH network, with the same parameter range as in the

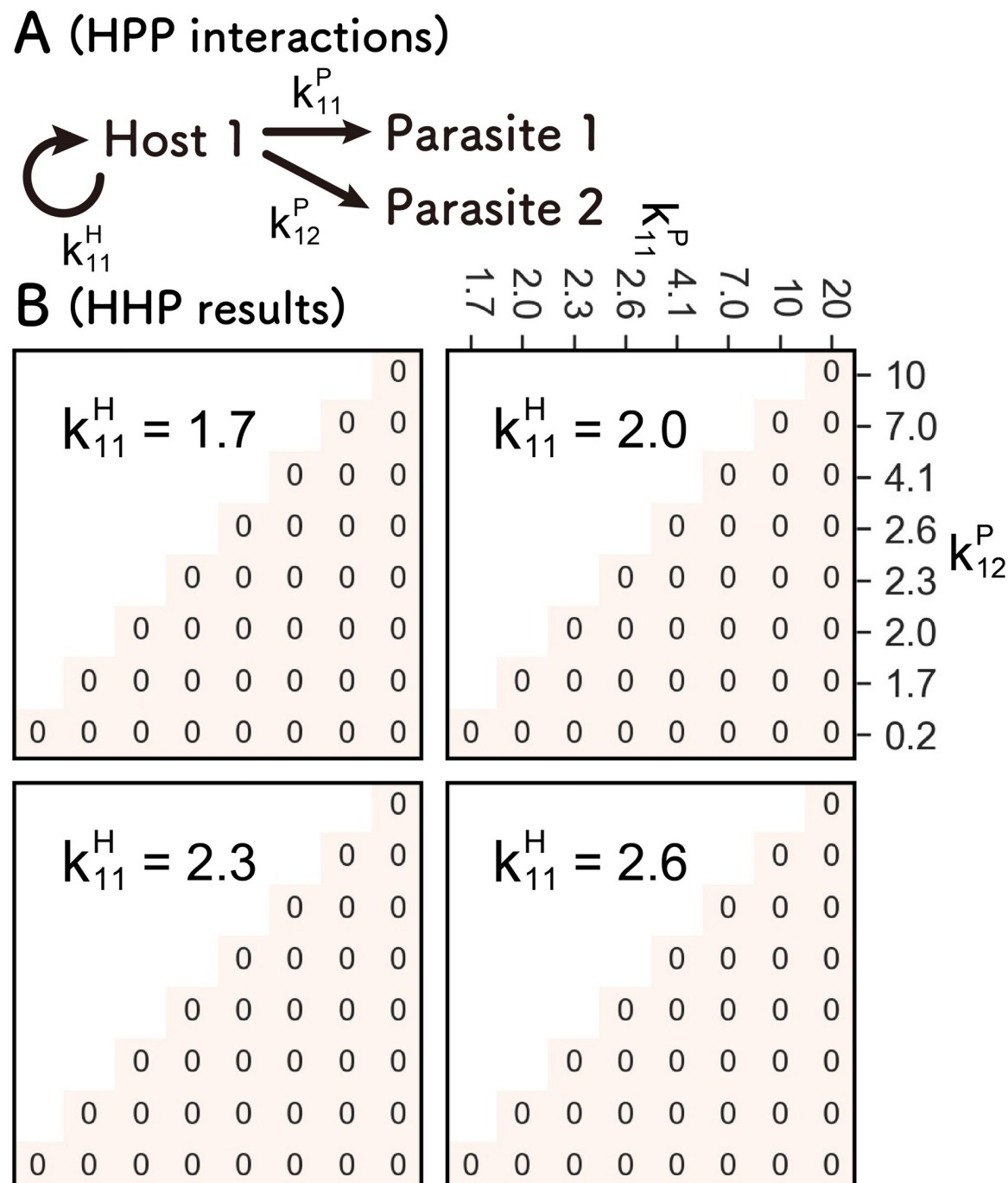

**Fig 4. Search for the parameters that allow sustainable HPP network.** (A) Scheme of the HPP network. The host self-replicates with coefficient $k_{11}^H$ and replicates two parasites with coefficient $k_{11}^P$ or $k_{12}^P$. (B) Numbers of the runs in which all three replicators (the host and Parasites 1 and 2) are sustained for 100 rounds out of 100 independent simulations.

HH network. Since there were too many parameter combinations to compute in a reasonable time, we used only two types of parasite coefficients: the same coefficient for both hosts ($k_{11}^P$ and $k_{21}^P = 7.0$, Fig 5A), termed "symmetrical parasite replication," where both hosts replicate the parasite similarly, or much smaller coefficient for one of the hosts ($k_{11}^P = 7.0$ and $k_{21}^P = 0.1$,

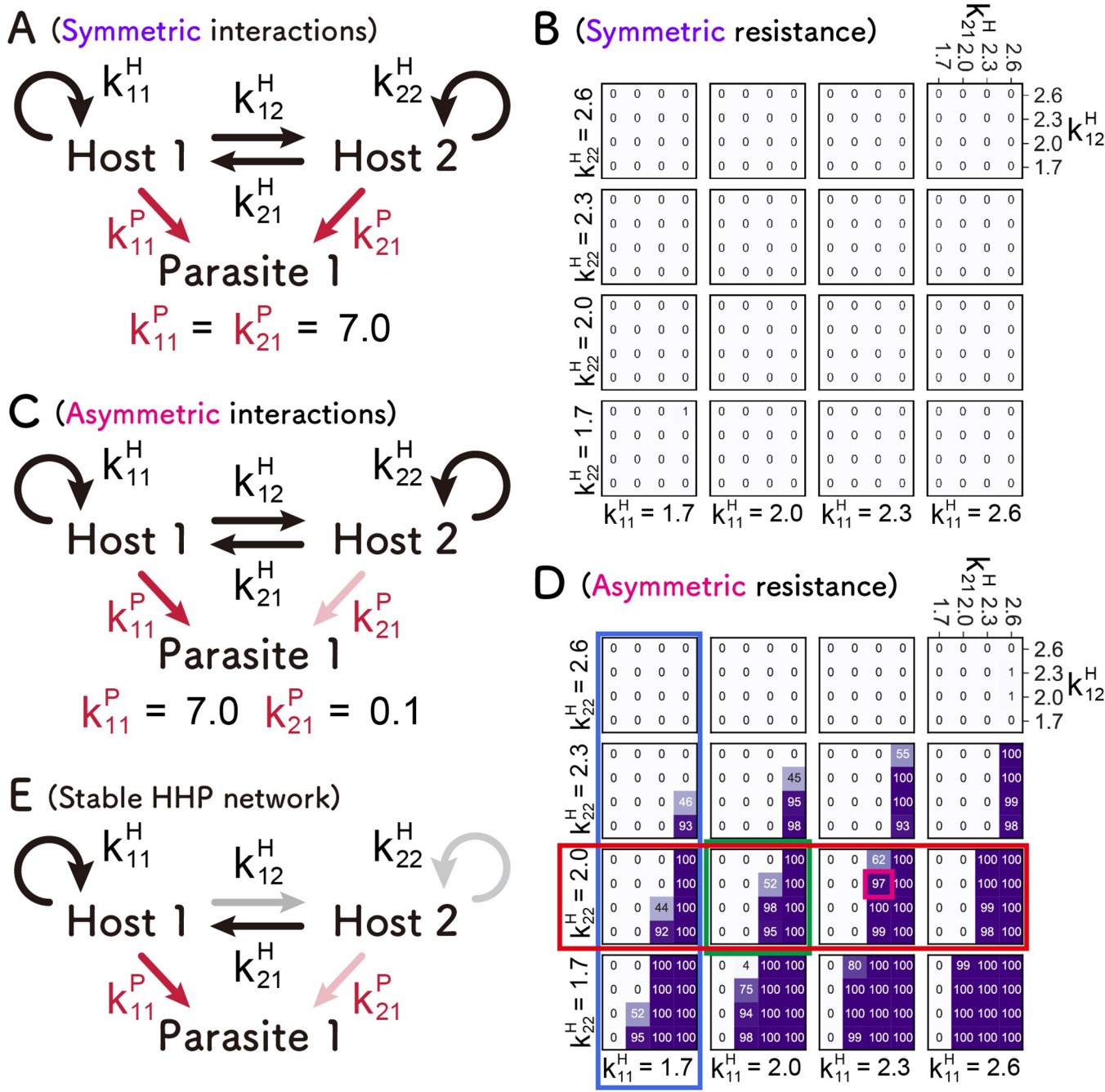

**Fig 5. Search for the parameters that allow sustainable HHP networks.** Symmetric (A) and asymmetric (C) HHP networks. The number of runs in which all three replicators (Hosts 1 and 2, and Parasite 1) were sustained for 100 rounds out of 100 independent simulations in symmetric (B) and asymmetric (D) cases is shown. The magenta square represents the close parameter values of the representative RNAs obtained from the evolutionary experiment. (E) A typical condition for a sustainable HHP network, which contains a parasite-susceptible and parasite-resistant host species; the parasite-susceptible host (Host 1) tends to replicate more efficiently through self- and/or cross-replications. The color depth of the arrows represents the value of the replication coefficients.

Fig 5C), termed "asymmetrical parasite replication," where one of the hosts is resistant to the parasite. The value (7.0) was adopted from the experimental data, and the value (0.1) was chosen as an example of much smaller values. We also simulated the case with an intermediate value of 1.0, but the results were similar (S3 Fig).

The number of sustainable replications out of 100 independent runs was strikingly different between the symmetrical and asymmetrical cases; in the symmetrical case (Fig 5B), sustainable replication was rarely observed with the parameter sets we tested. This is due to the competition between the two hosts; when the two hosts have the same susceptibility to the parasite, a host that replicates more competitively excluded the other host. By contrast, there were many parameter sets that allow sustainable replications, termed "sustainable parameters," in the asymmetrical case (Fig 5D). We also obtained a similar result with the extreme parameters (S4 Fig). These results indicate that the HHP network can be sustainable with a certain range of parameters when parasite resistance is asymmetrical between the two hosts. Furthermore, it should be noted that the sustainable parameters in the HHP network overlap with those in the HP network (i.e., the sustainable Host 1 and parasite in the HHP network are also sustainable in the HP network), which implies that a sustainable HP network can form a sustainable HHP network soon after the appearance of a parasite-resistant host. These results suggest that the transition from HP to HHP networks is a plausible pathway for complexification in the replication network.

We examined the asymmetric case in more detail. First, we found that as the self-replication of Host 1 (non-resistant host) increased (i.e., $k_{11}^H$ increased), the number of sustainable runs gradually increased. For example, in the parameter region of $k_{22}^H = 2.0$ (red rectangle in Fig 5D), the region with more than 90 number of surviving simulations increased from left to right (in the direction of increasing $k_{11}^H$). By contrast, as the self-replication of Host 2 (resistant host) increased (i.e., $k_{22}^H$ increased), the number of sustainable runs decreased. For example, in the region of $k_{11}^H = 1.7$ (blue rectangle), the region with 100 number of surviving simulations significantly decreased from bottom to top (in the direction of increasing $k_{22}^H$). Second, we found that as the cross-replication of Host 2 to Host 1 ($k_{21}^H$) increased, the number of sustainable runs increased. For example, in the region of $k_{11}^H = 2.0$ and $k_{22}^H = 2.0$ (green rectangle), the region with 100 number of surviving simulations increases from left to right (in the direction of increasing $k_{21}^H$). In summary, as a rough trend, a sustainable asymmetric HHP network requires parameter sets that favor replication of the parasite-susceptible host either by self-replication or cross-replication (a typical condition is schematically depicted in Fig 5E).

## HHH network

We investigated the HHH network (Fig 6A). Because there are too many parameters to simulate in realistic time in this network, we fixed the parameter values for Hosts 1 and 2 ($k_{11}^H$, $k_{21}^H$, $k_{12}^H$, and $k_{22}^H$) in two cases that allow sustainable replication in the HH network (Fig 3B): one of the Conditions I (i.e., "low self- and high cross-replications" conditions) ($k_{11}^H = 1.7$, $k_{21}^H = 2.6$, $k_{12}^H = 2.0$, and $k_{22}^H = 1.7$) and one of the Conditions II (i.e., "balanced replications" conditions) ($k_{11}^H = 2.0$, $k_{21}^H = 1.7$, $k_{12}^H = 2.0$, and $k_{22}^H = 1.7$). In these two cases, we searched for sustainable parameter sets for the new Host 3. All combinations of four values (1.7, 2.0, 2.3, and 2.6) were tested for the four new replication coefficients ($k_{31}^H$, $k_{32}^H$, $k_{13}^H$, and $k_{23}^H$) generated by the addition of Host 3. Only the smallest and largest values (1.7 and 2.6) were used for $k_{33}^H$, the self-replication coefficient of Host 3.

Under the "low self- and high cross-replications" conditions (Fig 6B and 6C), sustainable parameter sets were frequently found when $k_{33}^H$ is the smaller value, 1.7 (Fig 6B), whereas rarely found when $k_{33}^H$ is the larger value, 2.6 (Fig 6C), indicating that low self-replication also for the new Host 3, which induces inter-dependent replication of all replicators, is important for the sustainability.

A similar trend was also found under the "balanced replication" conditions (Fig 6D and 6E), the three hosts sustainably replicated with a certain range of parameter space with the

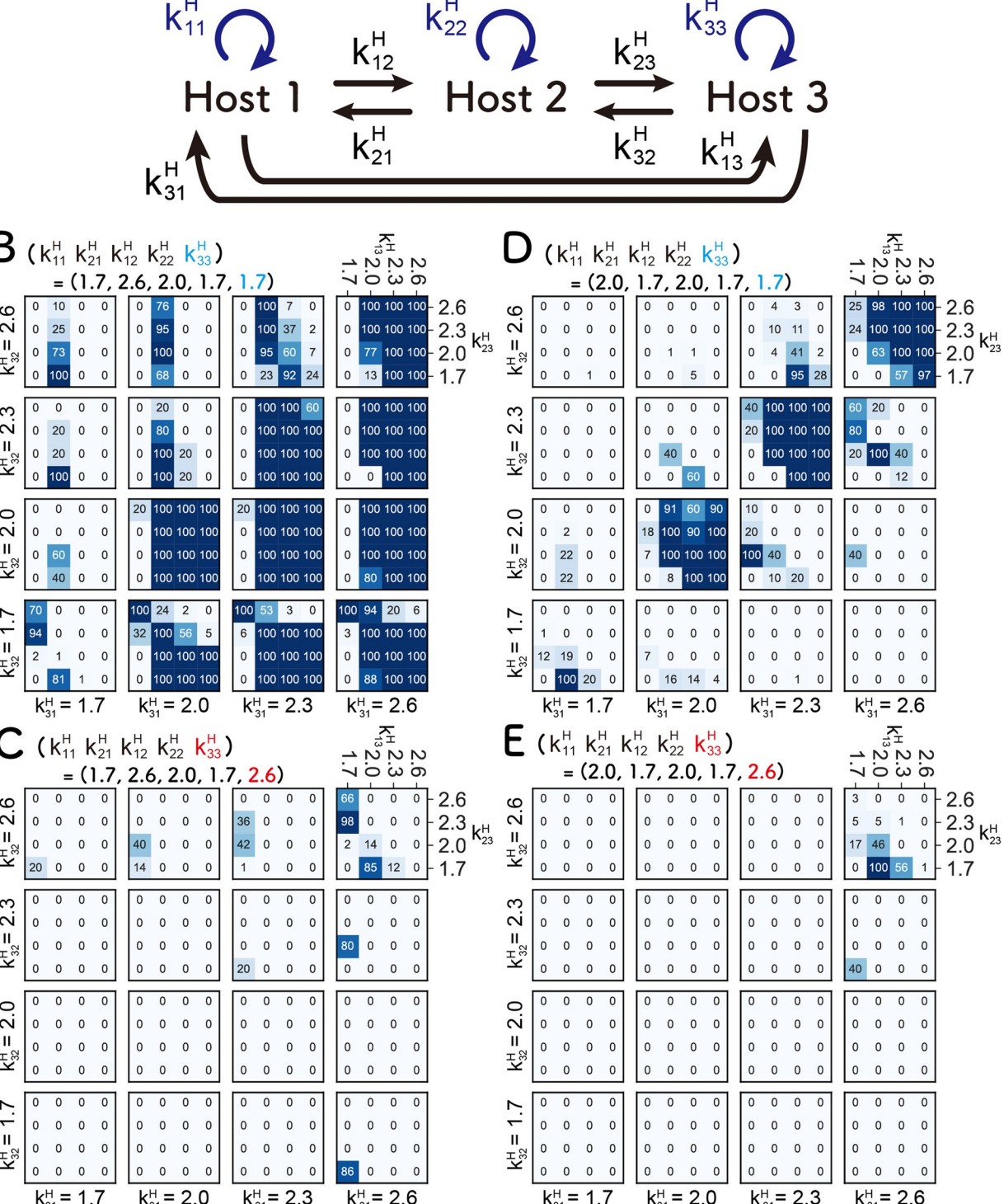

**Fig 6. Search for the parameters that allow sustainable HHH networks.** (A) Scheme of the HHH networks. (B-E) Numbers of runs in which all three hosts are sustained for 100 rounds out of three independent simulations. The parameter values for Hosts 1 and 2 are fixed at two cases that allows sustainable replication in the HH network. (B, C) Conditions I (Fig 3C, $k_{11}^H = 1.7$, $k_{21}^H = 2.6$, $k_{12}^H = 2.0$, and $k_{22}^H = 1.7$). (D, E) Conditions II (Fig 3D, $k_{11}^H = 2.0$, $k_{21}^H = 1.7$, $k_{12}^H = 2.0$, and $k_{22}^H = 1.7$). For the same reason, we employed a small (1.7) (B and D) or a large (2.6) (C and E) value for the self-replication coefficient of the newly added Host 3 ($k_{33}^H$).

smaller $k_{33}^H$ values (Fig 6D), whereas such parameter sets were rarely found with the larger $k_{33}^H$ values (Fig 6E). We found that when the parameters for Host 3 are also "balanced" (i.e., $k_{31}^H = k_{32}^H$), the HHH network tended to be sustainable under the condition.

In summary, when we added another host to the sustainable HH network, the resultant HHH network could be sustainable again with certain parameter sets, implying that host-only networks are plausible even when members of the network increase. However, a large obstacle for the formation of HH and HHH networks is the appearance of parasitic replicators, which is inevitable, at least in our experimental model. This point is further discussed in the discussion section.

In all aforementioned analyses, we counted only the numbers of sustainable runs in which all initial members of the network existed after 100 rounds. We also counted the number of runs in which a part of the replicators existed after 100 rounds. The data are shown in S5–S13 Figs for all networks.

## Computer simulation of transition of networks

Next, we investigated the possible evolutionary transitions of networks using computer simulations. We introduced a mutation step into the serial replication cycle immediately before the replication step. During the mutation step, a new host or parasite appeared at a certain rate in one of the compartments if the total number of replicator species in the system was less than three. A new host was generated from existing hosts at a rate of 0.02 per replication, and a new parasite was generated from hosts and parasites at rates of 0.001 and 0.002 per replication, respectively. These generation rates were partially based on experimental data (see Methods for details). A new host or parasite had replication coefficients randomly chosen from 1 to 3 or 0 to 10 as a type of double-precision floating-point number, respectively.

Starting with a single host species ($k_{11}^H = 2.0$), we performed 1000 rounds of serial replication cycles 1000 times. We counted the number of networks that were maintained for more than 100 rounds during the replication cycles and found that HPP, HHP, and HHH networks were maintained in 19, 218, and 14 runs, respectively (Fig 7A). This finding indicated that the HHP network was easier to maintain than other three-member networks, which was consistent with

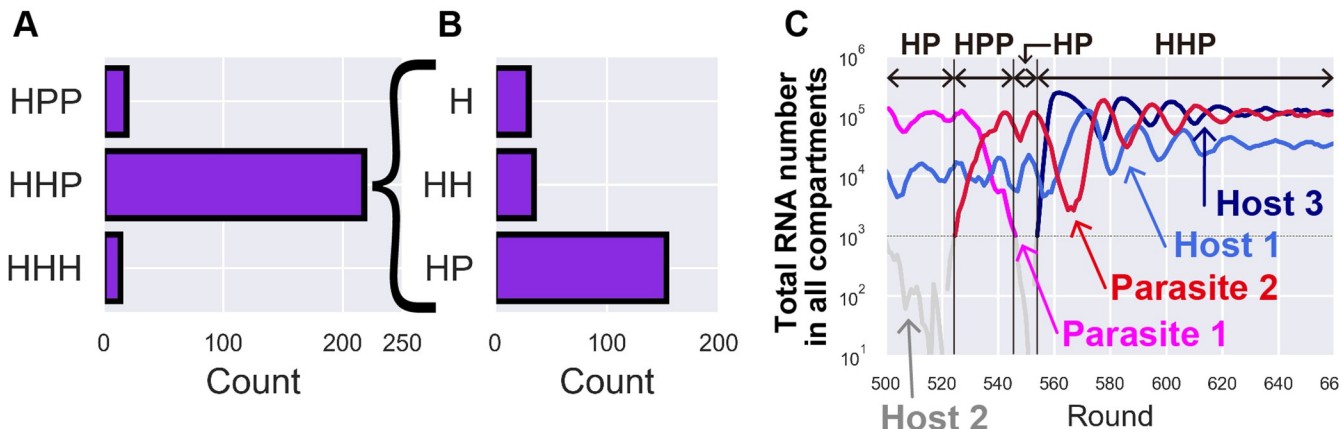

**Fig 7. Computer simulation of the evolutionary transition of replication networks.** The evolutionary transition was simulated by introducing a mutagenesis step in the serial replication cycle, as shown in Fig 1, for 1000 rounds. (A) The number of networks maintained for more than 100 rounds during replication cycles in 1000 simulations was counted. (B) Number of networks that preceded the 218 HHP networks shown in A. (C) A typical trajectory of the total number of each replicator in one of the simulations that resulted in the formation of the HHP network. The network composition was determined based on the replicators with more than 1,000. The replicators with less than 1,000 were shown as gray lines. The reason why some replicators started from >100 was that they replicated from 1 to >100 in the round that it appeared.

our sustainability analysis (Figs 4–6). To investigate the network from which the HHP network was formed, we examined the network immediately preceding the 218 HHP networks. Most (153) of the HHP network was preceded by HP networks (Fig 7B), supporting the hypothesis that the transition from HP to HHP network is the most plausible route. Fig 7C shows a typical example of the population dynamics in the simulation that reached the HHP network. An HP network was formed around round 550, and then it became an HHP network around round 560.

Next, we investigated the coefficients of 218 HHP networks maintained in the evolutionary simulation. According to the sustainability analysis shown in Fig 5, as a rough trend, the HHP network was sustainable when the parameters satisfied three conditions: 1) two hosts showed asymmetric resistance to the parasite (i.e., $k_{11}^{P} > k_{21}^{P}$), 2) the resistant host replicates the non-resistant host more than itself (i.e., $k_{21}^{H} > k_{22}^{H}$), and 3) the non-resistant host replicates equal to or more than resistant hosts (i.e., $k_{11}^{H} + k_{21}^{H} \geq k_{12}^{H} + k_{22}^{H}$). We found that 148 out of 218 HHP networks satisfied this condition (the parameter values are shown in S1 Data). This consistency between evolutionary-determined and stable parameter values has been suggested previously to support the reliability of sustainable conditions [34].

During 1000 rounds of replication cycles, all replicators disappeared in 796 of 1000 simulations. This high rate of disappearance was seemingly contradictory to the experimental results, in which we have never experienced disappearance of all RNA species [31]. This contradiction can be explained by the differences in the number of compartments. In this evolutionary simulation, we used 3000 compartments, whereas the experiment had approximately $10^{8}$ compartments. If the number of compartments decreases, the chance of accidental extinction of RNA species should increase. To verify the effect of the compartment number, we performed the same evolutionary simulation with smaller (1000) and larger (5000) number of compartments, in which the disappearance rate increased (936 disappearance in 1000 simulations) or decreased (470 disappearance in 1000 simulations), respectively. This finding supports that the smaller survival ratio in the simulation can be explained by the smaller number of compartments.

## Host and parasitic RNAs that may form HP and HHP networks in the previous evolutionary experiment

In our previous serial replication experiments of compartmentalized translation-coupled RNA replication, we found that a parasitic RNA appeared soon after starting replication and co-replicated with the original host RNA [29]. During further serial replication cycles, the host RNA diversified into two distinct lineages [30]. These results suggest that sustainable HP and HHP networks might have been formed during the evolutionary experiment, which is consistent with the simulation results. To confirm this possibility, we tested whether the dominant host and parasitic RNAs that appeared during the experiment had replication parameters that support sustainable HP and HHP networks.

To isolate dominant RNAs that possibly form HP and HHP networks, we analyzed the sequence data obtained in a previous study [30]. We focused on the early period (up to 39 rounds), where the host RNA starts to diversify. We chose the top eight most frequent sequences of each of the RNA populations in these rounds and drew phylogenetic trees for both the host and parasitic RNAs (Fig 8A and 8B), along with heat maps that represent the frequencies of each sequence (Fig 8C and 8D). For parasitic RNA, the phylogenetic tree (Fig 8A) and the frequency (Fig 8C) did not show any clear trends, but the most dominant parasitic RNA at round 13 remained as one of the dominant sequences until round 33. We chose this RNA (indicated as Parasite$^{1}_{\text{exp}}$) as representative of the parasite.

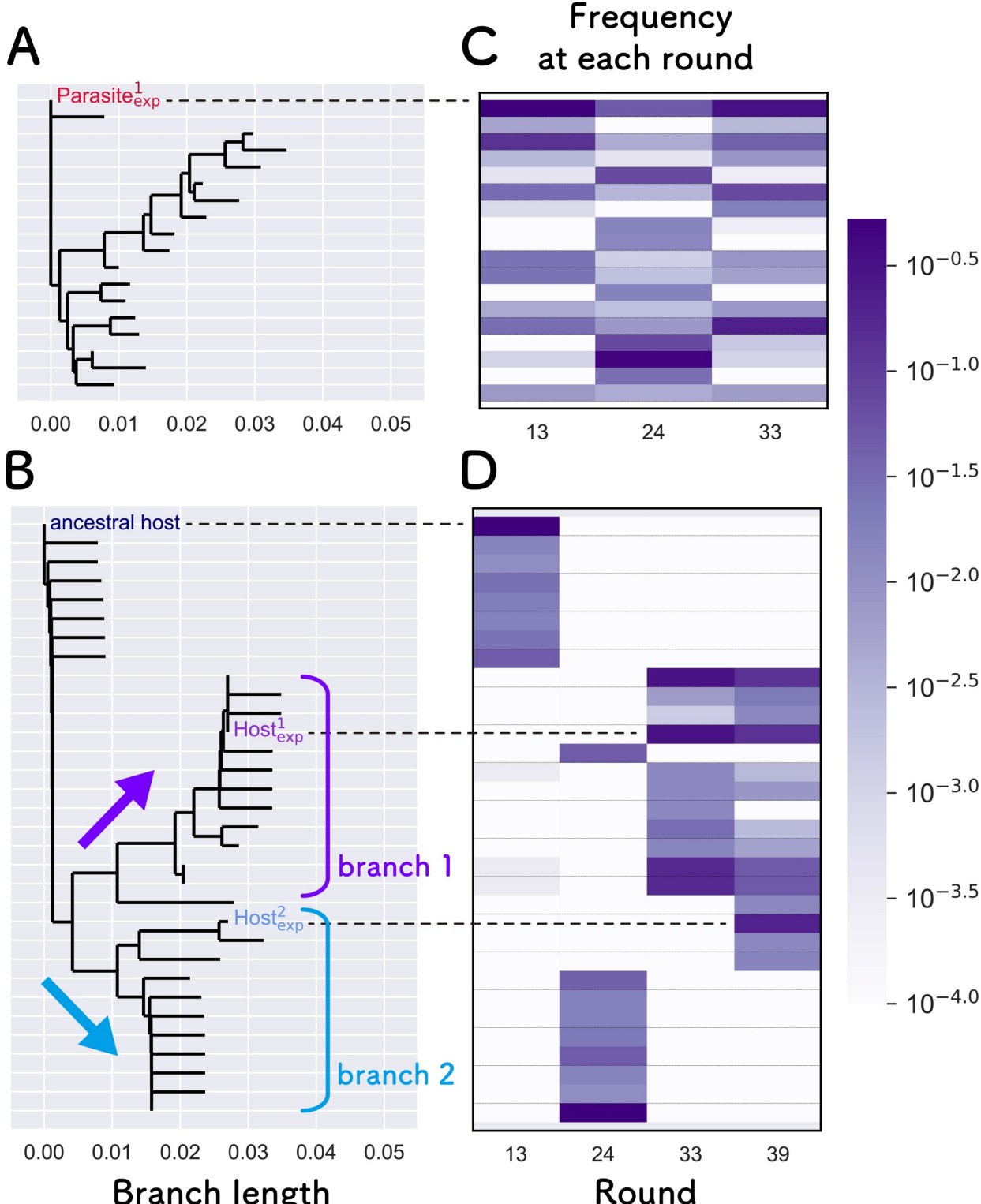

**Fig 8. Phylogenetic analysis of the host and parasitic RNAs that appeared in the previous evolutionary experiment.** Phylogenetic trees of the top eight parasitic (A) and host RNAs (B) that appeared in the early rounds of the previous evolutionary experiment [30]. Phylogenetic trees were constructed using the neighbor-joining method with the Phylo.TreeConstruction module in the Biopython library and default parameters [50–52]. The RNA frequencies at each round are shown as heat maps for the parasite (C) and host RNAs (D). Representative parasites and hosts used for the next biochemical experiments are indicated by "Parasite$^1_{exp}$" and "Host$^1_{exp}$"and "Host$^2_{exp}$," respectively. We could not obtain sequence

data of the parasite at round 39 because the total concentration of the parasitic RNA was too low. The horizontal scale of the phylogenetic tree is the same as the number of mutations.

The phylogenetic tree of the host RNAs was divided into two major branches (Fig 8B), consistent with the result of a previous study [30]. The frequency of the host RNAs changed significantly in each round (Fig 8D). At round 13, the sequences around the ancestral host dominated the population, and then, a part of the RNAs in branch 2 dominated the population at round 24. At round 33, the major RNA population changed to branch 1. At round 39, most of the RNAs in branch 1 remained as a major population, but some RNAs in branch 2 participated in the population as new dominant RNAs. From these results, we hypothesized that the dominant host RNA in branch 1 and a parasitic RNA form the HP network at round 33, which then changed to HHP network at round 39 by the addition of another host RNA in branch 2. To verify this hypothesis, we chose two representative hosts, one of the most frequent RNAs in branch 1 from round 33 to 39 (indicated as Host$^1_{exp}$) and the most common RNA in branch 2 at round 39 (indicated as Host$^2_{exp}$). Host$^1_{exp}$ and Host$^2_{exp}$ are separated by a Hamming distance of 7. Parasite$^1_{exp}$ are separated from Host$^1_{exp}$ and Host$^2_{exp}$ by Hamming distances of 8 and 6, respectively, except for the large deletion.

Notably, other than the large branches between branches 1 and 2, several small branches were observed within branches 1 and 2 (Fig 8B) and also in the phylogenic tree of parasites (Fig 8A). These small branches may belong to the same quasispecies.

## Parameter estimation of the representative RNAs

Next, we estimated the replication coefficients ($k_{ij}^X$) of Host$^1_{exp}$, Host$^2_{exp}$, and Parasite$^1_{exp}$. To measure the coefficients, we performed two-step reactions for all RNA combinations (Fig 9A). In the first translation reaction, RNA replicase is translated from one of the host RNAs, and in the second replication reaction, the translated replicase was used for replication of the same host and/or another host or parasitic RNA. The results of replication are shown in Fig 9B. The results indicated that parasite replication was asymmetric. When comparing the red bars, we found that the parasite was replicated when Host$^1_{exp}$ was used as RNA I (i.e., by Host$^1_{exp}$'s replicase), while it was barely replicated when Host$^2_{exp}$ was used as RNA I (i.e., by Host$^2_{exp}$'s replicase), indicating that Host$^2_{exp}$ is more resistant to the parasite, consistent with the sustainable asymmetric case shown in Fig 5C and 5D. To quantitatively compare the parameters, we estimated the replication coefficients from the replication results (Table 1). The replication coefficients of Host$^1_{exp}$ and the parasite are close to one of the sustainable conditions in the HP network (a light blue square in Fig 3F) and on the edge of the sustainable conditions in the HHP network (a magenta square in Fig 5D). These results indicate that the RNA species that appeared during the evolutionary experiment have properties that allow sustainable HP and HHP networks.

Note that we measured replication coefficients with clonal RNAs in this experiment, whereas during the evolutionary experiment, the RNAs replicated in a population comprised various RNA species. In a population, RNAs may be replicated with higher-order interactions among various RNAs. However, we analyzed the contributions of such higher-order interactions in our previous study and found that the contributions were minor, at least until round 240 (Supplementary Text 2, S17 and S18 Figs in Mizuuchi et al) [31].

We further tested whether the representative host and parasite RNAs co-replicated sustainably using compartmentalized serial replication experiments. We mixed the three representative RNAs at an equivalent concentration (10 nM) in a cell-free translation solution and

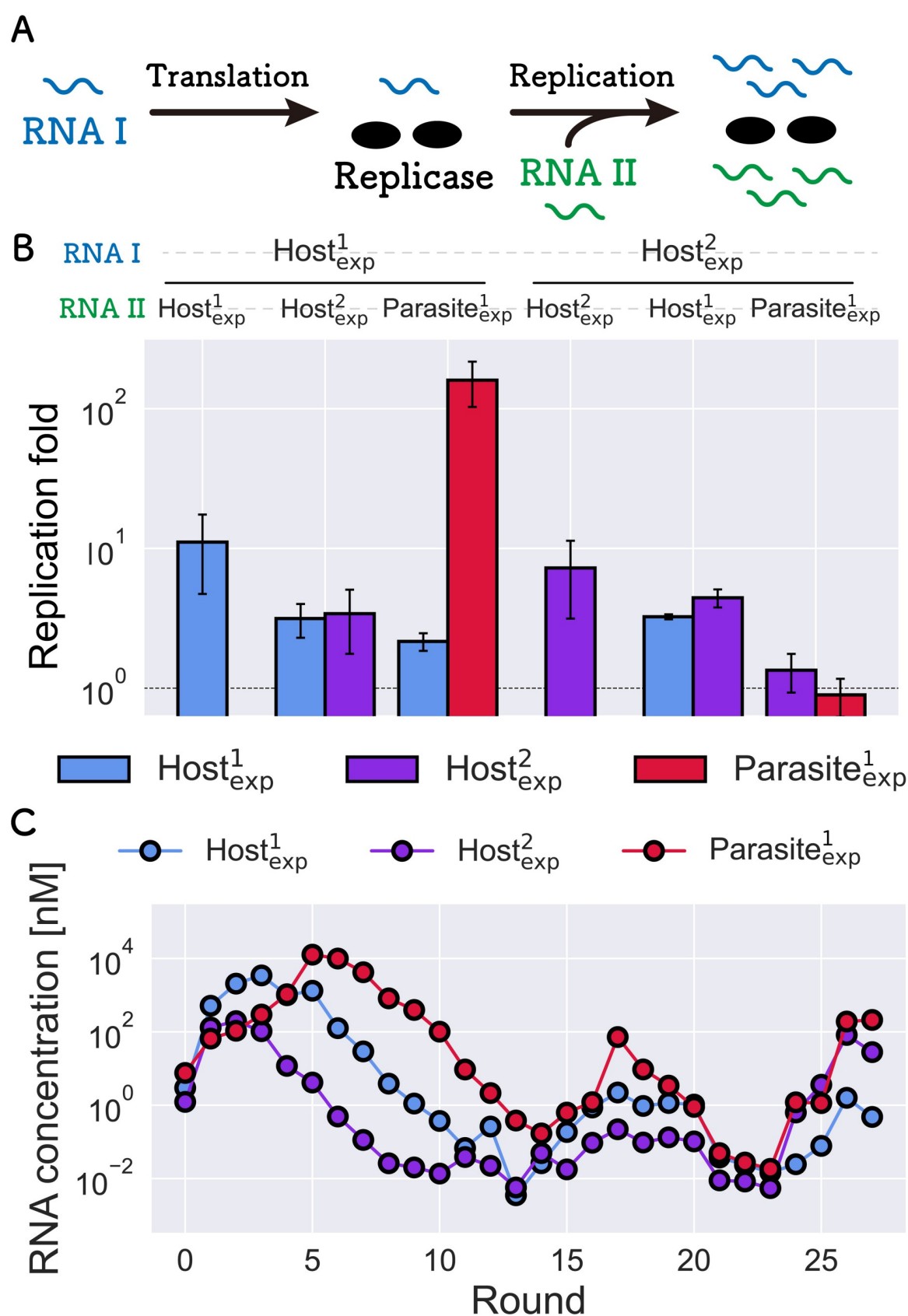

**Fig 9. Biochemical analysis of the representative host and parasitic RNAs.** (A) Experimental procedure for the estimation of replication coefficients. In the first translation reaction, RNA replicase was translated from one of the host RNAs (RNA I) for 2 h at 37°C, in which UTP was omitted to avoid RNA replication. In the second replication step, another host or parasitic RNA (RNA II), UTP, and an inhibitor of translation (30 μg/ml streptomycin) were added, and both RNAs I and II were replicated by the replicase for 1 h at 37°C. (B) RNA replication results. Experiments are independently performed three times. The error bars represent standard deviations. (C) Trajectory of RNA concentrations in the compartmentalized serial replication experiment of the three representative RNAs.

encapsulated them into water-in-oil droplets. The replication was repeated using the same serial replication procedure as in a previous study [30]. All three RNAs were replicated until 27 rounds while maintaining detectable concentrations (Fig 9C), supporting the notion that the selected host and parasite RNAs have the ability to form a sustainable HHP network.

## Discussion

In this study, we investigated the plausible complexification pathway of host-parasite replication networks using computer simulation and experiments. First, we examined the parameter space that allowed sustainable replication of all members in replication networks from two-(HH and HP) to three-member networks (HHH, HHP, and HPP). Sustainable parameter spaces are broader in HP and HHP networks for the range of the parameters we used, suggesting the plausibility of complexification from a single replicator to HP and then to HHP networks. We further confirmed that the dominant RNAs isolated from the previous evolutionary experiments had parameter sets that sustained HP and HHP networks, suggesting that the transition of replication network occurred during the evolutionary experiment, consistent with the previously-proposed scenario by Takeuchi and Hogeweg [16]. These results provide both theoretical and experimental evidence that support the validity of the previously proposed complexification scenario by Takeuchi and Hogeweg [16] and also provide evidence that the spontaneous development of a complex reaction network through Darwinian evolution is feasible within the parameter space that is achievable with RNA and proteins and that coevolution with parasitic replicators plays an important role in the complexification.

This study focused on the issue of how replicator networks evolve in terms of their complexity, which was similar to the previous theoretical study by Takeuchi and Hogeweg [16]. Details of the theoretical and experimental models used here are different from that presented previously. For replication, the model by Takeuchi and Hogeweg (TH model) assumes complex formation among RNA replicators based on their secondary structures and base pair matching, whereas in our experimental system, the RNA replication is catalyzed by a protein enzyme translated from an RNA replicator, and the interaction between the enzyme and RNA replicator is primarily determined by the template specificity of the encoded replication enzyme. Although working molecules are different, a common point for both TH and our experimental models is that the RNA sequence (via translation in our experiment) plays a central role in determining which RNA is replicated. Our theoretical model does not assume any underlying mechanism for the template specificity but only assigns replication coefficients. For spatial structure, the TH model assumes a two-dimensional square grid, and one grid contains at most one RNA molecule, which can interact with RNA molecules in the adjacent grids. In our theoretical and experimental models, we used compartment structures, which can contain many RNA molecules. All RNAs in the same compartment can interact with each other. For the propagation of RNAs, the TH model assumes diffusion of the replicated RNAs into adjacent grids. In our theoretical and experimental models, RNAs disperse into different compartments via occasional random fusion and division among any of the compartments in the system. Considering that a similar complexification process occurred in both the TH and

our models, these differences in the replication, spatial structure, and diffusion process are not critical factors for the evolution of complex replicator networks.

We used a very simplified model in the evolutionary simulation conducted in Fig 7, which allows for only three types of replicators at a time to save computational costs and does not include the trade-off between the replication coefficients because of the lack of knowledge of the mechanism underlying the trade-off. Owing to these limitations, the result of the simulation should be carefully interpreted. For example, the HHP network was maintained in 218 of 1000 runs in the simulation, but this number might be overestimated. With the limitation of the three types of replicators, once a stable HHP network appeared, no new replicator could invade the network. However, in a more realistic case without such a limitation, a new replicator that has a faster self-replication parameter might invade and destabilize the network. Furthermore, the probability of the appearance of such a faster replicator should depend on the trade-off between the coefficients. To perform a more realistic evolutionary simulation, further effort to minimize the computational cost and understand the trade-off between parameters would be needed.

In this study, we proposed one of the possible processes of complexification in a host-parasite replicator system. In the process, replication networks evolve by successively acquiring new RNAs from a single RNA to an HP network, followed by an HHP network. However, an alternative view for the complexification process is also possible, in which quasispecies play an important role. A quasispecies comprises a steady-state population of mutationally inter-connected genotypes. The mutation rate associated with RNA replication by Qβ replicates is high (approximately $9.1 \times 10^{-6}$ per nucleotide) [35], corresponding to approximately 0.02 mutations during every replication of host RNA. The RNA population was associated with a large quasispecies from the early stage of the evolutionary experiment, and the members of the quasispecies might have formed a replication network, such as an HHP network, from the beginning. In this alternative view, various replication networks, such as an HHP network, existed in the early stage, and some stable networks dominated the population in the later rounds. To understand the validity and contributions of these different complexification processes, further biochemical analyses of many RNAs included in the quasispecies would be needed.

To date, the conditions required for the coexistence of multiple replicators have been studied using various theoretical models [16, 36–39]. The sustainable conditions observed in this study are consistent with those of previous studies. For example, the sustainability of an HHP network that requires a parasite and asymmetric parasite resistance between the two hosts (Fig 5C) is consistent with the idea that the parasites play a role as a "niche" to sustain different types of host species [16, 38, 39]. In addition, the sustainability of HH networks that require larger cross-replication than self-replication (Fig 3C) is not a new concept because it is fundamentally the same as the cooperative relationship found in hypercyclic networks [18]. This consistency with previous theoretical studies, however, does not diminish the importance of this study because the novelty of this study is not to provide a new concept for the coexistence theory but to provide experimental evidence that the pathways and parameters can be realized by the action of biologically relevant molecules, such as RNA and proteins. In this study, we found that the RNAs and the encoded replicase protein were able to have replication parameters that permit sustainable HP and HHP networks under compartmentalized conditions. We also found that experimentally obtained RNAs are on the edge of the sustainable parameter space (shown in the magenta square in Fig 5D), which implies that a slight change in one of the parameters easily destroys the sustainability. The analysis of this study using an experimental model and relevant computer simulation revealed the realistic yet fragile nature of molecular replication networks.

We found that the HHHH network can be sustainable in certain parameter spaces (Fig 6B), suggesting that replication networks consisting of only host species can be another feasible complexification pathway. Such a replication network requires smaller self-replication and larger cross-replication values, and thus, it is similar to the hypercycles proposed by Eigen [18]. However, such replication networks might not last long because parasitic replicators would appear soon. Parasitic replicators are reported to be inevitable in self-replicators with a certain level of complexity [40] as shown by the appearance of parasitic replicators soon after the initiation of replication in our translation-coupled RNA or DNA replication systems [29, 41]. Once parasitic replicators appear, the HH network changes to a more sustainable HHP network. Therefore, the pathway from HH to HHHH networks is possible, but can be realized in limited replication systems where parasitic replicators rarely appear.

The importance of parasitic entities in diversifying host species through evolutionary arms race and its inevitability have been proposed in various organisms [40, 42–44], digital organisms [14], and molecular replicators [16, 30, 31]. Recent theoretical study supports the coevolution with parasitic entity expand the host's complexity [45, 46]. The relatively broader parameter space that allows sustainable HHP network may imply that HHP network, in which the newly appeared host uses the parasite as a "niche," is a reasonable consequence of coevolution between host and parasite. If this pathway continues, the network may further develop by acquiring a new parasite and then a new resistant host continuously (S14 Fig). The analysis of a more complex replication network that includes a larger number of replicators is a remaining challenge. Indeed, we recently reported that after 240 rounds of serial replication cycles, a five-member network that consisted of three hosts and two parasites appeared, in which the host RNAs have asymmetric resistance to the two parasites [31]. The following points are of utmost importance: how many types of RNAs participate in the next network, what determines the maximum number of members in a network, and whether the RNA members in a network eventually fuse to become a single molecule that encodes more information, which might lead to the origin of a multicistronic RNA genome [47]. The theoretical and experimental models used here provide a useful tool for answering these questions.

## Materials and methods

### Simulation of compartmentalized replication through serial replication cycle

The compartmentalized replication cycle consists of repeating three steps: replication, culling, and fusion-division (Fig 1A). In the replication step (Fig 1B), hosts and parasitic replicators in each compartment replicate depending on their numbers in each compartment according to the differential equations described below. The replication reactions are described using the following logistic equations that take self- and cross-replications among host and parasites into account:

$$\frac{\mathrm{d}H_i}{\mathrm{d}t} = H_i\left(\sum_j k_{ji}^H H_j\right)\left(1 - \frac{\sum_j H_j + \sum_h P_h}{N}\right), \qquad \text{Eq1}$$

$$\frac{\mathrm{d}P_h}{\mathrm{d}t} = P_h\left(\sum_j k_{jh}^P H_j\right)\left(1 - \frac{\sum_j H_j + \sum_k P_k}{N}\right), \qquad \text{Eq2}$$

where $H$ and $P$ are the number in a compartment of the hosts and parasites, respectively. $k_{ji}^H$ is the coefficient of the reaction in which host $j$ replicates the host $i$. $k_{jh}^P$ is the coefficient of the reaction in which host $j$ replicates the parasite $h$. $N$ is the carrying capacity and is the same for

all compartments. In this equation, we assumed that the replication rate of each host or parasite depends on three factors: own number ($H_i$ or $P_i$), the sum of the host number multiplied by its replication ability, $\left(\sum_j k_{ji}^H H_j\right)$ or $\left(\sum_j k_{jh}^P H_j\right)$, which represent the total replication ability provided by host replicators in the compartment, and the effect of carrying capacity in the compartment $\left(1 - \frac{\sum_j H_j + \sum_i P_i}{N}\right)$. This model does not include RNA degradation because it is negligible in the experimental system [30]. Compartments are assumed to be independent reactors, and there is no interaction between replicators in different compartments. The total number of compartments are fixed as $C$. The sizes of all the compartments were the same.

In the culling phase (Fig 1C), a certain number ($C_S$) of compartments was randomly selected, and $C−C_S$ empty compartments were supplied to maintain a fixed total number ($C$). The number of selected compartments ($C_S$) is defined as $C_S = \lfloor C \times S \rfloor$, where $S$ ($\in [0, 1]$) is the culling rate.

The fusion-division phase (Fig 1D) mimicked the experimental process to mix the contents of compartments through repeated fusion and division process of the compartment [30]. In this simulation process, the following three steps were repeated $A$ times ($A$ is defined as "fusion-division frequency"). First, two compartments were randomly chosen from all compartments. Second, the numbers of each replicator (i.e., hosts or parasites) in the two compartments were summed to mimic the compartment fusion. Third, the replicators in the fused compartment were randomly redistributed into two new compartments to mimic the division of the compartment according to the binomial distribution. In this step, stochasticity appeared in RNA composition, especially when the number of RNAs was small.

To search for parameter sets that allow sustainable replication, we conducted the replication-culling-fusion-division cycle for 100 rounds and counted each number of "sustained runs" out of 100 independent runs. We defined the "sustained run" as that where the number of all hosts or parasites in the network is greater than the number of compartments in the final round. In these simulations, all compartments were initially filled with equal numbers of all hosts and parasites, as much as the carrying capacity. The number of compartments ($C$) was 3,000, the culling rate ($S$) was 0.25, the fusion-division frequency ($A$) was 5,000, and the carrying capacity ($N$) was 100.

## Evolutionary simulation

A mutation step was introduced immediately before the replication step to simulate the evolutionary transition shown in Fig 7. In this step, a new host is generated from existing hosts at a rate of 0.02 per replication, based on the mutation rate of Qβ replicase [35] and the typical RNA size (2000 nt). A new parasite is generated from a parasite at a rate of 0.002 per replication, because of approximately 1/10-fold smaller RNA size. Since we could not find reliable data for the generation rate of a new parasite from a host, we used the value of 0.001, which was smaller than the generation rate of a new parasite from the parasite. To reduce the computational cost, the total number of replicator species was restricted to less than three. In other words, a new species could appear in the mutation step only when the total number of replicator species is one or two. For computational ease, only one type of new replicator can appear in one round for both the host and parasites (i.e., the simultaneous appearance of two types of hosts was not allowed, but the simultaneous appearance of one type of host and one type of parasite was allowed).

When the total number of replicator species was less than three, the mutation process was conducted as follows. First, the number of replicates at the replication step was counted for each replicator in each compartment. Second, the probability of the appearance of a new

replicator was calculated by multiplying the replication number with the mutation rate. If the probability defined for each replicator in each compartment was less than a random value $\in [0, 1]$, then a new host or parasite appeared in the compartment. The probability was typically very low, and thus, only one new host or parasite appeared in each round in most cases. However, for simplicity, if two new hosts or two new parasites simultaneously appeared, we assumed that the two new species were the same (i.e., had the same coefficients). Third, the coefficients of a new host (a new host $i$ replicated by host $j(k_{ji}^{H})$) were randomly chosen from 1 to 3 as a type of double-precision floating-point number. The coefficients of a new parasite $h$ replicated by host $j(k_{jh}^{P})$ were randomly chosen from 0 to 10 as a double-precision floating-point number. The simulation was started from a single host that self-replicated with a coefficient of 2.0 and continued for 1000 rounds of serial replication cycles. The number of compartments ($C$) was 3,000, the culling rate ($S$) was 0.25, the fusion-division frequency ($A$) was 5,000, and the carrying capacity ($N$) was 100. The code for the simulation has been uploaded to GitHub (https://github.com/Dokunuma/PrebioDivSim).

## RNA preparation

The representative host and parasitic RNAs were prepared by in vitro transcription using each plasmid as previously described [48]. The plasmids encoding each representative RNA (pUC_RK-Host-1, pUC_RK-Host-2, and pUC-RK-Parasite) were constructed in this study by introducing mutations into the plasmid pUC-N96 that encodes the original RNA by PCR with mutated primers. These mutations are listed in S1 Table. All RNA sequences are shown in the S1 Text.

## Replication experiments and estimation of the parameters

The procedure was based on a previous study [31], which included two steps. First, a host RNA (30 nM, RNA I) was incubated at 37˚C for 2 h in a cell-free translation system in which UTP was omitted to avoid RNA replication. The cell-free translation system is a reconstituted translation system of *Escherichia coli* [49]. The composition was customized and reported in a previous study [29]. Next, the initial reaction solution was diluted 3-fold in the cell-free translation system, which contains another RNA (10 nM, RNA II), 1.25 mM UTP, and 30 μg/mL streptomycin to inhibit further translation, followed by incubation at 37˚C for 1 h. The mixtures were diluted 10,000 fold with 1 mM EDTA (pH 8.0) and each RNA concentration was measured by quantitative PCR after reverse transcription using PrimeScript One Step RT-PCR Kit (TaKaRa, Japan) with specific primers (S2 Table). Reverse transcription was performed for 30 min at 42˚C, followed by 10 s at 95˚C. PCR was performed for 5 s at 95˚C and 30 s at 60˚C for 50 cycles.

To estimate replication coefficients, we first calculated the common logarithms of the increase ratios from 0 to 1 h as fold values, $v_{ij}^{h}$, where the subscripts $i$ and $j$ represent RNA species used as RNA I and II, respectively, and the superscript $h$ represents the measured RNA species ($i$ or $j$). The fold values when the same host RNA was used for both RNA I and II were utilized as the self-replication coefficients (i.e., $k_{ii}^{H} = v_{ii}^{i}$). The fold values when different RNAs were used for RNA I and II were utilized as the nonself-replication coefficients after normalization to eliminate the competition effect between RNA I and II on replication according to the following equation:

$$k_{ij}^{H\ or\ P} = k_{ii}^{H} \frac{v_{ij}^{j}}{v_{ij}^{i}}, \qquad\qquad \text{Eq3}$$

where we assumed that the ratio of the fold values of the competitive RNAs ($v_{ij}^{j}/v_{ij}^{i}$) is the same

as the ratio of the self-replication coefficient ($k_{ij}^{H\ or\ P}/k_{ii}^{H}$). The derivation of this equation was shown in the S1 Text.

## Compartmentalized serial replication experiment of the representative hosts and parasitic RNAs

The serial replication experiment shown in Fig 9C was performed according to a previous study [29]. Briefly, the initial reaction mixture contained 10 nM Host$^1_{exp}$, Host$^2_{exp}$, and Parasite$^1_{exp}$ in the reconstituted translation system described above. The solution (10 μL) was dispersed in 1 mL of the saturated oil phase with a homogenizer (Polytron Pt-1300d; Kinematica) at 16,000 rpm for 1 min on ice and incubated for 5 h at 37˚C. An aliquot (200 μL) of the droplets was diluted with 800 μL of the saturated oil phase, and a new solution of the reconstituted translation system was added. The solution was vigorously mixed with the homogenizer at 16,000 rpm for 1 min on ice and incubated for 5 h at 37˚C. Thus, we repeated the serial replication cycle for 27 rounds. After incubation, the droplets were diluted 100-fold with 1 mM EDTA (pH 8.0) and each RNA concentration was measured by quantitative PCR after reverse transcription using PrimeScript One Step RT-PCR Kit (TaKaRa) with each specific primer (S2 Table).

## Supporting information

**S1 Fig. Search for sustainable parameters in HH network with extreme parameters.** The simulation procedure was the same as that shown in Fig 3B except for using smaller (0.2) and larger (4.1) parameter values and a smaller number [10] of independent simulations.
(TIF)

**S2 Fig. Search for the parameters that allow sustainable HP, HH, and HPP networks with large numbers of compartments.** The number of compartments and the frequency of fusion-division were increased to 10,000 and 16,500, respectively. The number of runs in which all three replicators (Hosts 1 and 2, and the parasite) were sustained for 100 rounds out of 10 independent simulations are shown. (A) HP network. The replication coefficient for the host self-replication is fixed at 2.0 or 2.3. (B) HH network. (C) HPP network.
(TIF)

**S3 Fig. Search for the parameters that allow sustainable asymmetrical HHP network with intermediate $k_{21}^{P}$ values.** The simulations of the HHP network were conducted by the same method as Fig 5 except for employing an intermediate $k_{21}^{P}$ value (1.0). The number of runs in which all three replicators (Hosts 1 and 2, and Parasite 1) were sustained for 100 rounds in 10 independent simulations are shown.
(TIF)

**S4 Fig. Search for the parameters that allow sustainable HHP network with extreme parameters.** The simulations of the HHP network were conducted in the symmetrical (A) or asymmetrical cases (B) by the same method as Fig 5 except for employing extreme parameter values (0.2 and 4.1). The number of runs in which all three replicators (Hosts 1 and 2, and the parasite) were sustained for 100 rounds in 10 independent simulations are shown.
(TIF)

**S5 Fig. Remaining replicators after 100 rounds for HH networks.** Simulations were conducted as described in Fig 3B for 10 times.
(TIF)

**S6 Fig. Remaining replicators after 100 rounds for HP networks.** Simulations were conducted as described in Fig 3F for 10 times.
(TIF)

**S7 Fig. Remaining replicators after 100 rounds for HPP networks.** Simulations were conducted as described in Fig 4B for 10 times.
(TIF)

**S8 Fig. Remaining replicators after 100 rounds for symmetric HHP networks.** Simulations were conducted as described in Fig 5B for 10 times.
(TIF)

**S9 Fig. Remaining replicators after 100 rounds for asymmetric HHP networks.** Simulations were conducted as described in Fig 5D for 10 times.
(TIF)

**S10 Fig. Remaining replicators after 100 rounds for HHH networks for Fig 6B.** Simulations were conducted as described in Fig 6B for 10 times.
(TIF)

**S11 Fig. Remaining replicators after 100 rounds for HHH networks for Fig 6C.** Simulations were conducted as described in Fig 6C for 10 times.
(TIF)

**S12 Fig. Remaining replicators after 100 rounds for HHH networks for Fig 6D.** Simulations were conducted as described in Fig 6D for 10 times.
(TIF)

**S13 Fig. Remaining replicators after 100 rounds for HHH networks for Fig 6E.** Simulations were conducted as described in Fig 6E for 10 times.
(TIF)

**S14 Fig. A hypothetical parasite-mediated complexification pathway in replication networks.**
(TIF)

**S1 Table. Mutations in the representative host RNAs.**
(XLSX)

**S2 Table. Primer sequences.**
(XLSX)

**S1 Text. Derivation of Eq 3 and RNA sequences.**
(DOCX)

**S1 Data. Parameter values of 218 HHP networks at round 100 of the evolutionary simulation.**
(CSV)

## Author Contributions

**Conceptualization:** Rikuto Kamiura, Norikazu Ichihashi.

**Data curation:** Rikuto Kamiura, Ryo Mizuuchi.

**Formal analysis:** Rikuto Kamiura.

**Funding acquisition:** Norikazu Ichihashi.

**Investigation:** Rikuto Kamiura.

**Methodology:** Rikuto Kamiura, Norikazu Ichihashi.

**Project administration:** Norikazu Ichihashi.

**Resources:** Ryo Mizuuchi.

**Software:** Rikuto Kamiura.

**Supervision:** Norikazu Ichihashi.

**Validation:** Rikuto Kamiura.

**Visualization:** Rikuto Kamiura.

**Writing – original draft:** Rikuto Kamiura.

**Writing – review & editing:** Rikuto Kamiura, Ryo Mizuuchi, Norikazu Ichihashi.

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
