## [Decision Letter · Decision Letter 0]

10 May 2022

Dear Dr Ichihashi,

Thank you very much for submitting your manuscript "Plausible pathway for a host–parasite molecular replication network to increase its complexity through Darwinian evolution" for consideration at PLOS Computational Biology.

As with all papers reviewed by the journal, your manuscript was reviewed by members of the editorial board and by several independent reviewers. In light of the reviews (below this email), we would like to invite the resubmission of a significantly-revised version that takes into account the reviewers' comments. Following the comments by Reviewer 1, please make sure that a revised version of the manuscript accurately positions your work with respect to Takeuchi and Hogeweg (2008).

We cannot make any decision about publication until we have seen the revised manuscript and your response to the reviewers' comments. Your revised manuscript is also likely to be sent to reviewers for further evaluation.

Sincerely,

Ricardo Martinez-Garcia

Associate Editor

PLOS Computational Biology

Ville Mustonen

Deputy Editor

PLOS Computational Biology

Reviewer's Responses to Questions

**Comments to the Authors:**

Reviewer #1: The manuscript by Kamiura et al. investigates the possible way in which a replicator system increases its complexity using a computational model that mimics an experimental system synthesised by the authors’ group. Their results suggest that the most plausible way in which complexity increases is as follows: a system starting with a single host replicator evolves to add one parasitic replicator, followed by the evolution of another host replicator that is resistant to the parasitic replicator.

Major comments:

I have concerns about the academic fairness and scientific novelty of the manuscript. I start with academic fairness. In the abstract, the manuscript states, “The analysis shows that the most plausible complexification pathway from a single host replicator is the addition of a parasitic replicator, followed by the addition of a new host replicator that is resistant to the parasite.” This pathway of complexification is identical to that demonstrated in Takeuchi and Hogeweg (2008), which is quoted below for convenience (in the quote, “catalysts” refer to host replicators):

‘Thus, the presence of catalysts entails a “niche” (ecological functionality) for parasites. In the current system, the C-catalyst creates such a niche, and this enables the evolution of the G-parasite. Moreover, once the C-catalyst-G-parasite organization is established, it creates yet another niche, i.e., a niche for a phenotype that can escape from the G-parasite. Acquiring such a phenotype, the A-catalyst evolves. Finally, the establishment of such an alternative catalyst, in turn, creates a niche for a phenotype that can parasitize this alternative catalyst. This can cause the evolution of the U-parasite. ... In summary, the results of this section demonstrated a chain reaction of niche generation and speciation in an emergent ecosystem.’ However, the manuscript does not acknowledge that Takeuchi and Hogeweg (2008) describes the same pathway for the evolution of complexity as reported in this study. Instead, the manuscript states, “Furthermore, recent theoretical studies conducted by Takeuchi and Hogeweg showed that parasitic replicators induced diversification of RNA-like replicators through evolutionary arms race in compartmentalized structures and allowed the formation of more complex inter-dependent molecular networks” (Line 74-77). The authors’ description of Takeuchi and Hogeweg (2008) makes it impossible for the reader to see that the pathway demonstrated by the authors’ model is the same as that demonstrated by the model of Takeuchi and Hogeweg (2008). I think this equivalence needs to be explicitly stated in the manuscript (as well as the difference, as I mention at the end of my major comments).

In addition, Takeuchi and Hogeweg (2008) is published more than ten years ago. I do not think such a study is usually considered “recent”, as the authors write in the manuscript.

Another related issue is in how the manuscript formulates the central question of this study. The manuscript states, “These experimental results support the idea that coevolution between host and parasitic replicators can drive diversification and complexification. However, it is still unknown how such complexification is possible and competitive exclusion among RNA species is circumvented” (Line 96-98). This question has been tackled by Takeuchi and Hogeweg (2008), which theoretically demonstrates a pathway for the evolution of complexity in an RNA replicator system and the importance of parasites and population structure for the evolutionary stability of ecological diversity. However, the manuscript introduces the above question as if no one has attempted to address it, which is wrong.

The authors might have written the above things because they had inadvertently missed the points of Takeuchi and Hogeweg (2008). However, just in case that is not the case, I would like to add the following. If one covers up previous studies to glorify one’s own studies, it will damage one’s academic credibility in the long term, even if one might get away with the short-term benefit of more publications.

The other major issue I have is that I do not see the scientific novelty of the modelling results as stated in the manuscript. I am not talking about the modelling results per se, but talking about the novelty as stated in the manuscript. In Lines 405-409, the manuscript states, “the novelty of this study is not to provide a new concept for the coexistence theory but to reveal realistic pathways and parameters for the complexification in biochemical replicator systems.” First, this quote indicates that the authors themselves do not think this study provides a new theoretical concept, which is in agreement with what I wrote above in relation to Takeuchi and Hogeweg (2008).

The second part of the above quote says that this study reveals more realistic pathways and parameters than known before. However, I do not see how this is achieved in this study. The authors’ model mimics a synthetic evolving system engineered by the authors. Mimicking a particular experimental system certainly makes it easier to test modelling results experimentally. However, in my opinion, that by itself does not necessarily make modelling results more realistic or relevant to the question of the origins of life than the other models because the level of abstraction employed in the authors’ model is similar to that of many other models, and there is not enough evidence indicating that the particular experimental system mimicks prebiotic reality.

The manuscript also states that a subset of the model parameters is based on experimental measurements. However, that does not necessarily make the model more realistic than the others either, for the following reasons. First, in Lines 165-169, the manuscript states that one of the two conditions in which two host replicators can coexist is that rate constants are balanced (k_11=k_12, k_21=k_22). The values of k_11, k_21 etc., were taken from experimentally measured values. However, it is evidently unrealistic that each pair of rate constants is exactly identical to one another, even if particular values of rate constants were obtained through experiments. There will be errors in the measurements too.

Another reason why using experimentally measured parameters does not necessarily make the model more realistic is that the model still uses multiple parameters that have not been experimentally measured, e.g., mutation rates and the ratio of mutating into parasites or hosts. Consequently, the results of the model do not necessarily match those of the experiments. For example, the manuscript states, “Starting from a single host species (k 11 = 2.0), we performed 3000 rounds of serial replication cycles for 100 times and found that in most of the runs (97 runs), all replicators were diluted out, while in three runs, the HHP network was formed.” Did the experiments also result in the evolution of the surviving network only in three out of 100 replicates? If so, is this fact explicitly reported, e.g., in Bansho et al. 2016 or Furubayashi et al. 2020?

To increase the plausibility of the modelling results, I think it is better to test a wide range of parameters rather than using a selected few or to let parameters be determined by evolution as suggested by Hogeweg (https://doi.org/10.1007/978-3-030-71737-7_2). In fact, the authors do test parameter values that appear to be selected without experimental measurements according to the manuscript. In my opinion, what makes this type of modelling results more realistic is evolutionary stability rather than setting a subset of parameters to experimentally measured values.

There is another reason that makes me think that the modelling results described in the manuscript are not novel. In Line 163, the manuscript reports one of the two conditions required for two host replicators to coexist (HH network), namely that two hosts replicate cooperatively. That cooperation is necessarily has been known since the 70s through the hypercycle theory of Eigen and Schuster. In Line 405, the manuscript states that the HH network is “similar” to the hypercycle. What is the difference? Given that the hypercycle can survive in a well-mixed system in the absence of parasites, one can easily expect that it can also survive in a loose-compartment system because the main effect of loose compartmentation is to make a system more resistant to parasites, but the system lacks parasites. So, in my opinion, there is hardly any novelty in this part of the authors’ results.

In my opinion, the novel results of this study are two folds. First, the authors’ model shows that the pathway of complexification proposed by Takeuchi and Hogweg (2008) is valid in a loosely-compartmentalised system without spatial self-organisation. Second, the authors’ experiments lend support to their modelling results. So, in my opinion, the manuscript is actually answering a different question from that mentioned above. The manuscript states, “One of the remaining challenges is the plausibility of such complexification within a realistic parameter space that is achievable with biologically relevant molecules such as RNA and proteins.” The manuscript tackles this challenge by theoretically generalising and experimentally supporting the pathway of complexification predicted in Takeuchi and Hogeweg (2008).

Overall, I cannot recommend this manuscript in its current state for PLOS Computational Biology or, in fact, for any other journals, because of its grave issues regarding academic fairness and the presentation of scientific novelty. I strongly urge a complete re-write of the manuscript to address the issues described above (and below).

Minor comments:

It is not explained how the equation in Line 541 comes about. Please describe the mathematical justification.

The replication rate was measured under experimental conditions that were different from those in which evolutionary experiments were done. Specifically, a solution contained only one or two genotypes in the measurements, whereas a solution contained multiple genotypes in evolutionary experiments. Could you discuss whether this could potentially make any difference? For example, the activity of enzymes could be modulated by the presence of other RNA molecules? Could there be RNA-RNA interactions?

In Line 503-505, the manuscript says, “The coefficients of a new host are randomly chosen from 1.7, 1.8, 1.9, 2.0, 2.1, 2.2, 2.3, 2.4, 2.5, and 2.6. Coefficients of a new parasite were randomly chosen from 0.01, 0.1, 0.5, 1, 2, 3, 4, 5, 6, 7, 8, 9, and 10.” I did not understand exactly what the authors did. Do the coefficients refer to k_ji values? If so, parasites do not replicate anything, so they should not have positive coefficients. Moreover, there are nine distinct k_ji values if the system contains three genotypes (i and j can each take three values, so three times three). How is each of the k_ji values selected from a set of values listed above?

The equations in Lines 459-460 assume continuous variables. However, I believe that the system is unstable without any stochasticity in the distribution of molecules over compartments. Something is missing in the description of the model, probably around Lines 479-481.

In Lines 420-423, the manuscript appears to suggest that the HH network (hypercycle) cannot survive in the presence of parasites in a loose-compartment system. However, as shown by Boerijst and Hogeweg (1991 Physica D), population structure can enable a hypercycle to survive in the presence of parasites. I feel that this result can be extended to a loose-compartment system because the authors have experimentally shown that two cooperating hosts can survive in a loosely-compartmentalised system even if mutations can generate parasites (Mizuuchi et al., 2022 Nat Comm). Thus, if the authors’ model and experiments are consistent with each other, the model is expected to display the survival of a hypercycle in the presence of mutations into parasites. If not, there is something more to learn about.

Minor errors:

Line 232: “allows” should be “allow”.

Line 424: (39) should be (38).

Line 489: I do not understand this sentence, “all compartments contain all hosts or parasites on average”.

Reviewer #2: Dear Authors:

Please find my comments in the attached document.

Best regards.

**Have the authors made all data and (if applicable) computational code underlying the findings in their manuscript fully available?**

Reviewer #1: **No: **The source code of the model is not provided.

Reviewer #2: **No: **I cannot certify that all data has been provided. This manuscript relies on several earlier papers, and it got very confusing tracking what is obtained from where.

PLOS authors have the option to publish the peer review history of their article (what does this mean?). If published, this will include your full peer review and any attached files.

Reviewer #1: No

Reviewer #2: No
---

## [Decision Letter · Decision Letter 1]

16 Sep 2022

Dear Dr Ichihashi,

Thank you very much for submitting your manuscript "Plausible pathway for a host–parasite molecular replication network to increase its complexity through Darwinian evolution" for consideration at PLOS Computational Biology.

As with all papers reviewed by the journal, your manuscript was reviewed by members of the editorial board and by three independent reviewers. Two of the reviewers are the same that reviewed the original submission. Both of them appreciate your efforts in addressing all their previous comments and list a series of remaining issues that should be addressed. Considering some of the comments raised in the first round of revision, I invited a new external reviewer, whose comments are also attached to this letter.

In light of the three reviews, we would like to invite the resubmission of a significantly-revised version that takes into account the reviewers' comments. We cannot make any decision about publication until we have seen the revised manuscript and your response to the reviewers' comments. Your revised manuscript is also likely to be sent to reviewers for further evaluation.

Sincerely,

Ricardo Martinez-Garcia

Academic Editor

PLOS Computational Biology

Ville Mustonen

Section Editor

PLOS Computational Biology

Reviewer's Responses to Questions

**Comments to the Authors:**

Reviewer #1: I thank the authors for their revision. The revision has addressed the issues I raised with the positioning of the authors' work in the context of existing work.

Reading the whole manuscript again, I noticed a few additional points that appear to be important enough to raise, even though this is the second review.

Lines 98-107 contain duplicated sentences.

Line 279-281 says, 'in the parameter region of k^H_22=2.0 (red rectangle in Fig 5D), the region with 100 points increased from left to right (in the direction of increasing k^H_11).' However, this statement does not appear to be correct, according to Figure 5D. The figure seems to show that the number of regions with 100 points barely changes with k^H_11: 8 regions (k^H_11=1.7), 8 regions (k^H_11=2.0), 7 regions (k^H_11=2.3), and 8 regions (k^H_11=2.6). What actually happens is that the range of k^H_12 for which HHP persists when k^H_21=2.0 shifts upwards as k^H_11 increases.

I am not sure if the statement in Lines 288-289 is accurate. The statement says, 'In summary, a sustainable asymmetric HHP network requires parameter sets that favor replication of the parasite-susceptible host either by self-replication or cross-replication.' This statement could be interpreted as saying that the survival of HHP is facilitated by increasing k^H_11 or k^H_21. Interpreted this way, the statement does not appear to be true. Figure 5D shows that the survival probability of HHP is 0.96 when k^H_11=2.3, k^H_12=2.0, k^H_21=2.0, and k^H_22=2.0. Compared to this parameter set, increasing k^H_11 to 2.6 decreases the chance of survival to 0.08, which contradicts the above interpretation of the statement.

Similar to the above statement, the sentence in Line 358 says, "the HHP network was sustainable when ... non-resistant hosts replicated more than resistant hosts (i.e., k^P_11>k^P_21 and k^H_11+k^H_21 > k^H_12+k^H_22).' The inequality implies that the survival of HHP is enhanced by increasing k^H_11 or k^H_21 or decreasing k^H_12 or k^H_22. This proposition does not seem to be correct either. Figure 5D shows that the survival probability of HHP is 0.96 when k^H_11=2.3, k^H_12=2.0, k^H_21=2.0, and k^H_22=2.0. Compared to this parameter set, decreasing k^H_12 to 1.7 decreases the probability of survival to 0.07. The above results, together with the previous paragraph, appear to suggest that a sustainable HHP network requires parameter sets for which H1 is sufficiently fitter than H2 in the absence of P, but not too fit to drive H2 to extinction before P increases to give H2 an advantage.

Finally, the mutation in the authors' model is restricted such that there can be at most three "species" of replicators in the system. This restriction appears unrealistic, especially in view of the fact that the novelty of the model is said to lie in the availability of the corresponding experimental system, where mutations are not restricted. The assumed restriction on mutations raises an important question: are the modelling results robust if mutations are allowed to occur irrespective of the number of species? I suggest that the authors discuss this question because the assumed restriction seems quite strong. If mutations are allowed to occur irrespective of the number of species, host replicators might evolve towards maximising their self-replication coefficients, resulting in a situation where k^H_11 and k^H22 are nearly equal to each other. In this case, HHP would not be able to persist because H1 could not out-compete H2 in the absence of P. Likewise, parasitic replicators might evolve towards maximising both k^P_11 and k^P_21. In this case, HHP would not persist because H1 is not more susceptible to P than H2 is.

The following is an additional comment related to the question posed above. The authors' model does not incorporate any genotype-phenotype map and assumes that k^H_22 mutates in the same way as k^H_11 does (i.e., random walks between 1 and 3). By contrast, in the authors' experimental system, the genotype-phenotype map is likely to be complex, and the kinetic parameters of replicators of different "species" certainly do not mutate in the same way. The genotype-phenotype map might play a role in the evolution of HHP because it can impose various trade-offs, e.g., between being resistant to parasites (i.e., decreasing k^P_21) and maximising the self-replication rate (i.e., increasing k^H_22) for host replicators, or between parasitising H1 (i.e., increasing k^P_11) and parasitising H2 (i.e., increasing k^P_21) for parasitic replicators. Such trade-offs, if they existed, could prevent H2 from evolving a replication coefficient as high as that of H1 and, similarly, prevent P from parasitising both H1 and H2, thereby allowing the persistence of HHP. Thus, a complex genotype-phenotype map and consequent trade-offs might play a role in the evolution of HHP in the experimental system, which is not taken into account in the authors' model.

Reviewer #2: Dear Authors, dear Editor:

Please, find my report in the attached document.

Reviewer #3: Revision of manuscript:

Plausible pathway for a host-parasite molecular replication network to increase its

complexity through Darwinian evolution

Authors:

Rikuto Kamiura, Ryo Mizuuchi, and Norikazu Ichihashi

The manuscript by Kamiura and colleagues connects the experimental results from the Ichihashi group to the mathematical/computational theory of the RNA world.

To do this, they construct their own mathematical model that mimics the well-known experimental system developed in the Ichihashi group. The model consists of a population of protocell-like compartments inhabited by replicating RNAs and possibly parasites. Replicators and parasites grow within compartments, and the compartments undergo fusion, fission, and sampling to introduce new resources. The model, partially parametrised with their experimental data, is used to study the population dynamical stability of different combinations of interacting replicators and parasite species. The authors show that some networks of replicators and parasites can stably coexist. They also identify a plausible route for the evolutionary complexification of the system: parasites evolve to exploit replicators, and new replicator lineages evolve that escape the parasites. Interestingly, this complexification through niche construction turns out to be the same as in an independently derived model by Takeuchi & Hogeweg (2008). The authors further study the long-term stability of their results with evolutionary simulations. Finally, they check the experimental validity of their conclusions by re-analising previous data and running a short experiment with a three species system.

I agree with one of the reviewers that the earlier theoretical results from Takeuchi & Hogeweg (hereafter TH) had to be made more central in the manuscript, as they turned out to be supported by the authors’ model and confirmed by their experiments. To their merit, the authors clearly acknowledge this in the revised manuscript.

It happens rarely in Origin of Life research that an independently derived model is so strikingly confirmed by experiments – I think this is worthy of publication. Moreover, the authors study a model which is slightly different from TH. While the model is a lot simpler than TH, it is close to the authors’ experiments, which provides the “bridge” for the comparison between the authors’ experimental work and the theory developed by TH.

This being said, I had some trouble following the manuscript. I see that the paper has already undergone revision, but I have some general points that I feel should be addressed, plus there are a few parts that really confused me.

General comments:

1) About the paragraph: “Strategy of theoretical model and analysis”. I found the first paragraph impossible to understand without first having read how the model is constructed. Please explain the model first, and then what you plan to do with it. I think it is fine that the authors put the model in relation to the literature on modelling the RNA world, but as it stands now the model is not explained enough to allow understanding the results obtained with it. So I recommend expanding the model explanation section – perhaps going along Fig. 2 (which could be Fig. 1).

Also, am I correct that the RNAs are assumed to be very stable – so there is no decay in the model, and compartments that reach carrying capacity just halt all replication occurring within them (and only compartments merged with empty ones carry on replication)? If so, this should be explained somewhere, as it is the reason why parasites do not take over every compartment in which they are initially present. I wonder if this is also the case in the experimental system.

2) Do I understand correctly that 100 instances of the HH network are run each for 100 rounds? If so, I find this problematic. As far as I can tell (though I might be wrong) the only difference between two such instances is some randomness in the composition of the compartments. So if e.g. 20 runs out of 100 survive after 100 time steps – how many survive after 200 time steps? After 1000? The point here is: is long-term coexistence stable whenever a small number of runs survive to 100 time steps? I suspect not, but either way this should be explicitly stated/explained. Also, the plots do not clearly discriminate the runs that go extinct (if any) and those in which the original complexity is lost and a simpler network of species remains.

3) I think the authors sufficiently frame their results in the context of the theory from TH. However, the differences between models (and between TH model and the authors’ experiments) should be also emphasised. A detailed comparison would highlight what is general and what is specific to the models, and would make for a more insightful discussion.

Detailed comments:

Line 47: you mean “evolves” here, not “develops”, right?

Line 93: The sentence mentions hypercycles, but reference 22 is about proto-chromosomes (linked replicators). Is this reference in the right place?

95: “… studies have revealed that spatial structures such as compartmentalization repress …”

The sentence sounds like compartmentalisation is a subset of spatial structure, but it should be the other way around.

101: please change “developed” to “evolved” in the sentence about ecological complexity.

Also, the term “by successive addition” is incorrect. In Takeuchi & Hogeweg’s study, the parasite evolves spontaneously – exactly like in the author’s experimental system.

99 and 104: the term “compartmentalised structure” is used. Takeuchi & Hogeweg’s model is spatially structured. RNAs are implicitly compartmentalised because of local interactions – but I do not understand what the term means.

105 – 110: These two sentences are somewhat repeated from line 100 – 105. Something went wrong with the formatting?

131: the word “appearance” here should really be “evolution”.

135: I am not convinced by the word “generalised”. What makes your model more general? Wouldn’t you say that your model is analogous to Takeuchi & Hogeweg’s, but follows your experiment more closely?

155: I don’t understand the following: “... the parameter spaces that allow for the sustainable replication of all members in the networks for certain generations.”. Do you mean that you run the system’s dynamics for a fixed number of time steps?

163: In the sentence “The replication is continued by repeating three steps: replication, selection, and fusion-division”, the word “replication” appears twice, but it has two different meanings. Please use a different word (perhaps the first “replication” could be substituted with “Within-compartment dynamics”). Notice that this sentence re-appears in Material and Methods, where it should also be changed. Moreover, I find the word “selection” here a bit puzzling: aren’t you removing 25% of the compartments without specifically selecting for any property? Perhaps “culling” could be a better word.

197: a parenthesis is opened but not closed.

243: Sentence: “These results suggest that even if an HPP network is formed during evolution, it will soon return to the HP network.” This is interesting, but no data is shown to support this.

343-352 and Fig. 7: it is difficult to extract information from the figure. Fig. 7A is partially a bar chart (frequency of evolved networks) and partially numbers (presumably the frequency of ancestral networks of HHP). Please make the information homogeneous in the figure. In Fig. 7B, how can I infer that HHP descends from HP? Moreover, are the lines representing concentrations over one compartment or multiple ones? There is no caption/legend for the grey lines. Why are the initial concentrations of Host 2 and Parasite = 10^2?

358: about the sentence: “We found that 170 out of 218 HHP networks satisfied this condition”. It would be interesting to see the evolutionary trajectory (or at least the evolutionary steady states) of the various evolutionary parameters.

414: I do not understand the sentence “These small branches are expected to be derived from quasispecies”.

415: You state: “The interpretation of this result based on quasispecies is discussed later.”. But later, in the discussion, the results are not really discussed – you simply state that you do not expect Host 1 and 2 to be part of the same quasispecies (more on that below).

474: I do not understand the definition of quasispecies is as “a population of variants with random mutations”. A quasispecies is a (steady-state) population of mutationally inter-connected genotypes.

480: I do not understand what this sentence means: “The existence of many different but closely related genotypes can be explained by quasispecies”. Do you mean that you expect these closely related genotypes to be part of the same quasispecies?

481-484: You report Hamming distance = 7 between Host1 and Host2 as a way to argue that they belong to different quasispecies. This is a result and should be moved to the Results section. But aside from this, I do not understand how two sequences separated by 7 nucleotides out of 2041 (the length of Host 1) can be different quasispecies. Can you clarify why Host 1 and Host 2 are not variants within the same quasispecies (with Host 2 better able to escape the parasite)?

533: I do not understand what message this sentence is conveying: “It is of utmost importance how many RNAs participate in a network, what determines the maximum number, and whether the different RNAs fuse to become a single molecule that encodes more information, which may lead to the origin of chromosome (22).”. In particular, I do not understand the connection between RNA networks and evolution of chromosomes.

**Have the authors made all data and (if applicable) computational code underlying the findings in their manuscript fully available?**

Reviewer #1: Yes

Reviewer #2: Yes

Reviewer #3: Yes

PLOS authors have the option to publish the peer review history of their article (what does this mean?). If published, this will include your full peer review and any attached files.

Reviewer #1: No

Reviewer #2: No

Reviewer #3: **Yes: **Enrico Sandro Colizzi
---

## [Decision Letter · Decision Letter 2]

1 Nov 2022

Dear Dr Ichihashi,

Thank you very much for submitting your manuscript "Plausible pathway for a host–parasite molecular replication network to increase its complexity through Darwinian evolution" for consideration at PLOS Computational Biology. As with all papers reviewed by the journal, your manuscript was reviewed by members of the editorial board and by several independent reviewers. The reviewers appreciated the attention to an important topic. Based on the reviews, we are likely to accept this manuscript for publication, providing that you modify the manuscript according to the review recommendations. Please notice that Reviewer #3 has some few minor comments remaining.

Sincerely,

Ricardo Martinez-Garcia

Academic Editor

PLOS Computational Biology

Ville Mustonen

Section Editor

PLOS Computational Biology

Reviewer's Responses to Questions

**Comments to the Authors:**

Reviewer #1: I would like to comment on the issue of quasispecies, which was discussed by the other reviewers and authors. If I understand it correctly, the critical question appears to be whether Hosts 1 and 2 belong to the same quasispecies or not. If they belong to the same quasispecies, it appears illegitimate to model the populations of Hosts 1 and 2 as those of two distinct species, potentially undermining the authors' interpretation of their experimental results as embodied in their model, as suggested by another reviewer.

To address the above question, we need an empirical test to judge whether or not two genotypes belong to the same quasispecies. One possible test might be to check whether two genotypes are separated by a sufficiently large Hamming distance, but such a test is problematic because it is unclear how large the distance must be, as discussed by the other reviewers. A better test might be to examine whether two genotypes can coexist with each other without any mutation. If they can, the result is compatible with the interpretation that the two genotypes belong to separate quasispecies. The test is not definite proof, but it is a good start because it is simple. It is also in line with the original motivation behind the concept of quasispecies. In their seminal work on the hypercycle, Eigen and Schuster asked how macromolecules accumulate information through Darwinian evolution. They discovered that the amount of information one quasispecies can accumulate is bounded above (aka information threshold). So, they looked for a way to evolve multiple quasispecies in one system because that would increase the system's information capacity. Thus, they proposed a hypercycle, in which different genotypes coexist with each other, not by mutating into each other as in a quasispecies, but by catalysing each other's replication in a cyclic manner. Therefore, coexistence without mutation would support the existence of multiple quasispecies.

A test similar to the above has been, in fact, already performed by the authors, as depicted in Figure 9C. This figure shows that three genotypes sampled from Hosts 1 and 2 and Parasite can coexist without the initial presence of the other genotypes. The chance of mutations creating Host 1 from Host 2 (or vice versa) in 7 replication steps is about [0.02 * (1/2000)]^7 ~ 1e-35 (0.02 is the mutation rate per genome, and 2000 is the genome size). So, the mutational influx between Hosts 1 and 2 is negligible, at least for the first ten or so rounds of replication. Moreover, the authors might have data to check if any genotypes other than the initial three become prominent within the first 20 rounds of replication. If the initial three genotypes remain to be the majority during this time, it indicates that the mutational influx is negligible for the first cycle of oscillation. That would support the interpretation that Hosts 1 and 2 can survive without mutating into each other. A result like that might be the best evidence one could hope for the quasispecies issue until a better test is proposed in the future.

This is a follow-up to the above discussion. In response to a comment by another reviewer, the authors removed the paragraph discussing whether Hosts 1 and 2 belong to the same quasispecies based on the Hamming distance. Although I agree that the paragraph is not quite tenable, I also find it useful if the paper contains information about the Hamming distance between three representative genotypes of Hosts 1 and 2, and Parasite as well as their genome lengths in one location (I was not able to find this information in the other parts of the manuscript). So, I suggest that the authors put this information back. In addition, I suggest that the authors explain the meaning of the scale of branch lengths in the caption of Figure 8, as I was not able to find this information in the manuscript.

Overall, I think this manuscript makes an important contribution to the field. I agree with another reviewer saying that it is rare in this field to see that a theoretical prediction receives experimental support. I am of the opinion that this manuscript, combining both theoretical and empirical approaches, contains enough body of evidence to warrant publication. So, I recommend the publication of this manuscript in this journal.

Reviewer #3: I have a couple of very minor comments. After these comments are addressed, I do not need to review the paper again, and I am happy to recommend that the article is accepted.

Fig 1b: the second sum sign lacks subscripts.

Line 166, “in which” should be “with which”.

Line 176, “1,7” should be “1.7”, right?

Line 277, 280, 284, the word “point” is used? Can you please change this to “number of surviving simulations” or something similar?

Line 417, I do not understand what “branches derived from quasispecies” means. Do you mean (as I already asked in the previous round) that you expect that these genotypes belong to the same quasispecies? If not, could you describe what of quasispecies theory you are referring to?

To me, these short branches seem just a consequence of the evolutionary process: some mutants survive long enough for you to be able to sequence them – but I cannot see where the quasispecies part is.

**Have the authors made all data and (if applicable) computational code underlying the findings in their manuscript fully available?**

Reviewer #1: **No: **The Hamming distance between three representative genotypes of Hosts 1 and 2, and Parasite as well as their genome lengths. The meaning of the scale of branch lengths in Figure 8

Reviewer #3: Yes

PLOS authors have the option to publish the peer review history of their article (what does this mean?). If published, this will include your full peer review and any attached files.

Reviewer #1: No

Reviewer #3: No

Figure Files:

Data Requirements:

Reproducibility:

References:

---

## [Editor Report · Decision Letter 3]

4 Nov 2022

Dear Dr Ichihashi,

We are pleased to inform you that your manuscript 'Plausible pathway for a host–parasite molecular replication network to increase its complexity through Darwinian evolution' has been provisionally accepted for publication in PLOS Computational Biology.

Best regards,

Ricardo Martinez-Garcia

Academic Editor

PLOS Computational Biology

Ville Mustonen

Section Editor

PLOS Computational Biology

---

## [Editor Report · Acceptance letter]

8 Nov 2022

PCOMPBIOL-D-22-00138R3 

Plausible pathway for a host–parasite molecular replication network to increase its complexity through Darwinian evolution

Dear Dr Ichihashi,

I am pleased to inform you that your manuscript has been formally accepted for publication in PLOS Computational Biology. Your manuscript is now with our production department and you will be notified of the publication date in due course.

With kind regards,

Anita Estes
